# ENHANCING CONVERSATIONAL AGENTS WITH SKILL-OF-MIND-INFUSED LARGE LANGUAGE MODEL

## ABSTRACT

To foster social bonding, humans naturally develop the ability to select appropriate conversational skills (e.g., empathy) based on situational context—a cognitive process we term *skill-of-mind*. However, LLMs often struggle to generate human-like responses in complex social dialogues. To address this, we propose a 100K skill-of-mind-annotated conversation dataset, MULTIFACETED SKILL-OF-MIND, which includes 38 conversational skills across various interactive scenarios (e.g., chitchat), grounded in diverse social contexts (e.g., demographics). Using this dataset, we introduce a new family of skill-of-mind-infused LLMs, 🗿 THANOS, with model sizes of 1B, 3B, and 8B parameters. We also introduce a comprehensive benchmark suit, THANOSBENCH, for assessing both capabilities of skill-of-mind and response generation in LLMs. Through extensive experiments evaluating 12 LLMs, THANOS demonstrates performance comparable to Claude-3.5-Sonnet, even outperforming LLaMA-3.1-405B. Specifically, THANOS enhances LLM-generated responses, making them more human-favorable and empathetic communication. Because we find out that recent high-performing LLMs still struggle to exhibit superior skill-of-mind capabilities, we believe it is invaluable to highlight the inherent challenges in this area.

## 1 INTRODUCTION

In everyday conversations, humans engage in diverse and complex interactions with their interlocutors (e.g., friends, colleagues) by understanding and interpreting their interlocutors' situations Rashkin (2018); Lee et al. (2022a) and personas Zhang (2018); Lee et al. (2022b). Moreover, its interactions are progressed along with recalling memorable events or moments Bae et al. (2022a); Jang et al. (2023); Lee et al. (2024d). For example, as shown in Figure 1, humans reflect on which skill to use for the next turn by internally reasoning about which skill would be most appropriate. This process evolves through self-reflection and feedback, as people assess the positive or negative reactions of their interlocutors. We refer to this entire process as *skill-of-mind*, which involves interpreting and understanding the current dialogue situation, planning the best skill strategy for the next response, and then selecting the most appropriate conversational skill (e.g., empathy).

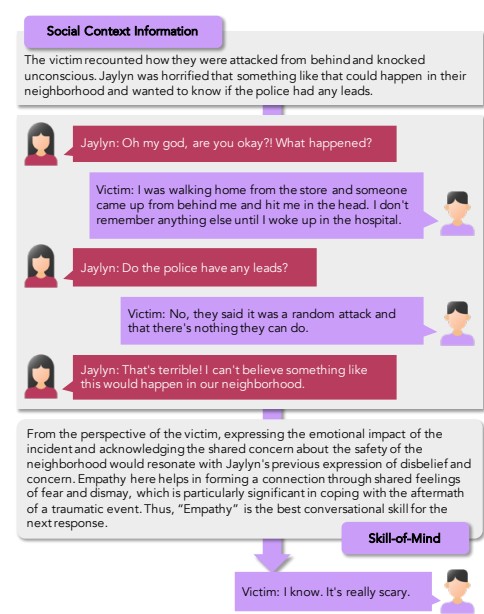

Figure 1: An overview of ***skill-of-mind*** process.

Recently, conversational agents powered by LLMs Touvron et al. (2023); AI@Meta (2024); Team et al. (2024b) have demonstrated impressive capabilities in logical reasoning Pan et al. (2023) and creativity Franceschelli & Musolesi (2023). However, they continue to struggle with social commonsense reasoning Chae et al. (2023) and strategic communication in interactive environments Zhou et al. (2023). We posit that directly generating the next response based on

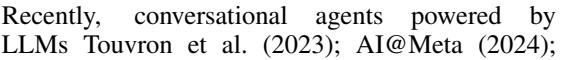

conversational skills is particularly challenging for LLMs due to the complexity of social dialogue. Specifically, there are (1) multiple plausible responses Li et al. (2015); Bao et al. (2019) or (2) multiple conversational skills Smith (2020); Yang et al. (2024b) in the world to respond a given situation. To address this challenge, we suggest that before generating a response, LLMs should first interpret the dialogue context and plan the most plausible conversational skill—similar to how human internal states guide communication. This approach can enhance response quality, even in dynamic social interactions, by providing a structured form of guidance.

In this work, we introduce MULTIFACETED SKILL-OF-MIND, a collection of multi-turn, multi-faceted *skill-of-mind* annotated conversations. This dataset includes annotations for both explanations and conversational skills, covering 38 conversational skills across one-sided turns within dialogues. The dataset is derived from 12 existing source dialogue datasets, which encompass diverse social contexts and scenarios (e.g., chitchat, counseling). To annotate *skill-of-mind*, we prompt GPT-4 (i.e., `gpt-4-turbo`) to generate explanations and identify conversational skills from a pre-defined collection of conversational skills. Using this dataset, we propose a new family of *skill-of-mind*-infused LLMs, 🐱 THANOS , which generate both an explanation and the most appropriate conversational skill, given the previous dialogue history and social context.

To rigorously assess LLMs in both skill-of-mind and response generation capabilities, we propose a holistic evaluation suite, THANOSBENCH, which comprises three datasets for the skill-of-mind task and five datasets for the response generation task. In total, we compare THANOS with 12 LLMs in THANOSBENCH across two different tasks. For the skill-of-mind task, THANOS-8B demonstrates competitive skill-of-mind capabilities, performing on par with Claude-3.5-Sonnet and outperforming LLaMA-3.1-70B-Instruct (+2.54) and LLaMA-3.1-405B-Instruct (+2.01), though it still lags behind GPT-4o and Qwen-2.5-72B-Instruct. However, we find that even the best-performing models achieve relatively low scores (17.04 in Qwen-2.5-72B-Instruct), highlighting the inherent difficulty of this task, even for state-of-the-art LLMs.

For the response generation task, we observe that THANOS-3B effectively guides LLMs to generate more natural and engaging responses, without requiring additional training, in pairwise comparison evaluations. Notably, THANOS-3B significantly enhances the response quality of GPT-4o (60.06% in naturalness) and Gemini-2.0-Flash (58.25% in naturalness). Furthermore, compared to common-sense reasoning inference, skill-of-mind plays a more effective role in improving response generation quality. Additionally, we find that THANOS promotes empathetic communication by aligning with human empathetic patterns. These findings highlight both the importance of skill-of-mind and the effectiveness of THANOS as a socially aware guidance mechanism.

Our contributions are summarized as follows: (1) We introduce a new social concept, *skill-of-mind*, which involves interpreting dialogue situations, planning the best skill strategy, and selecting the appropriate conversational skill. (2) We present a multi-turn, multifaceted skill-of-mind-annotated conversation dataset, MULTIFACETED SKILL-OF-MIND, which encompasses diverse social dynamics and interactive scenarios. (3) Using MULTIFACETED SKILL-OF-MIND, we propose a family of skill-of-mind-infused LLMs, THANOS, with model sizes of 1B, 3B, and 8B parameters. (4) For thorough evaluation, we introduce a holistic benchmark suit, THANOSBENCH, for assessing both skill-of-mind and response generation capabilities in LLMs. (5) Through extensive experiments, we demonstrate the effectiveness THANOS in THANOSBENCH.

## 2 MULTIFACETED SKILL-OF-MIND

### 2.1 PRELIMINARIES: SKILL-OF-MIND

**Motivation behind "Skill-of-Mind".** In the conversational AI literature Smith (2020); Kim et al. (2022c), these "skills" are sometimes viewed as communication strategies Zhou et al. (2023), but more broadly refer to a range of desirable abilities essential for maintaining continuous and meaningful dialogue with an interlocutor. Such skills span from general proficiencies (e.g., empathy, persona) to task-oriented functions (e.g., making phone calls, booking a hotel). During everyday social interactions—a critical component of human conversation Myllyniemi (1986)—people routinely endeavor to understand and interpret their interlocutors' beliefs and mental states. We regard this internal cognitive process as *theory of mind* Premack & Woodruff (1978), which also considers relevant aspects of the interlocutor (e.g., demographics, personal background, and relationship).

Grounding on this understanding, individuals select the most fitting conversational skill to respond effectively.

**Formulation of "Skill-of-Mind".** Let $D = \{(s_i, u_i)\}_{i=1}^{t-1}$ denote a dialogue, where each speaker $s_i \in \{A, B\}$ [1] and $u_i$ is the corresponding utterance. Let $ctx$ represent the social context. At turn $t$, the "skill-of-mind" is formalized as $f_{SoM} : (D, ctx) \mapsto (e_t, cs_t)$, where $e$ and $cs$ denote an explanation/rationale and conversational skill, respectively.

**Ingredients for "Skill-of-Mind".** The concept of "skill-of-mind" consists of three main components: (1) social context ($ctx$), (2) explanation/rationale ($e$), and (3) conversational skill ($cs$), which are described as follows.

- **Social Context Information:** Socially interactive dialogues involve a wide range of social dynamics, such as demographics, personal experiences, and relationships. We believe that these factors influence the *skill-of-mind*. For instance, emotional empathy is more appropriate in conversations with a significant romantic partner than with an AI teaching mentor. Therefore, when considering social context information, we take into account various elements, such as the situation, social relationships, persona, and memory. [2]

- **Explanation/Rationale:** This involves interpreting and understanding the current situation to determine the most optimized conversational skill for generating an engaging response that strengthens social rapport Zech & Rimé (2005) with the interlocutor in the given dialogue. To achieve a higher quality of explanation, we adopt *perspective-taking*-style Davis (1983); Ruby & Decety (2004), which prompts GPT-4 to imagine the actual speaker in the dialogue. The explanation is represented in free-form sentences.

- **Conversational Skill:** Real-world scenarios involve a diverse range of conversational skills—for example, empathy and persona management in chitchat, hotel reservation management in task-oriented dialogues, and memory recall in long-term conversations. To capture this diversity, we cover 38 conversational skills in MULTIFACETED SKILL-OF-MIND based on existing studies. A complete list of these skills, along with the corresponding studies that informed their selection, is provided in Appendix D.2.

## 2.2 DATASET CONSTRUCTION PROCESS

Based on the above ingredients of skill-of-mind (§ 2.1), we first collect source dialogue datasets and annotate them with skill-of-mind.

**Step 1: Source Dataset Collection.** To build more flexible and versatile skill-of-mind-infused LLM, we collect 12 multi-turn dialogue datasets, which are publicly available online: Soda Kim et al. (2022a), ConversationChronicles Jang et al. (2023), ProsocialDialog Kim et al. (2022b), EmpatheticDialogues Rashkin (2018), Wizard-of-Wikipedia Dinan et al. (2018), Cactus Lee et al. (2024b), Casino Chawla et al. (2021), MultiWOZ 2.2 Zang et al. (2020), PersuasionForGood Wang et al. (2019), Pearl Kim et al. (2024a), Syn-PersonaChat Jandaghi et al. (2023), and Stark Lee et al. (2024d). In total, we collect source dialogues from the training sets. We then split each dialogue into sub-dialogues by focusing on one-sided exchanges. For example, given a dialogue $\mathcal{D} = \{(s_i, u_i)\}_{i=1}^4$, we create two sub-dialogues: $\mathcal{D}_1 = \{(s_i, u_i)\}_{i=1}^2$ and $\mathcal{D}_2 = \{(s_i, u_i)\}_{i=1}^4$. We remove sub-dialogues with fewer than four turns, as we believe that early in the dialogue, there is a higher distribution of non-informative skills, such as greetings, rather than informative skills. We then randomly sample sub-dialogues from each source dataset in specific proportions. As a result, we obtain a total of 100K dialogues.

**Step 2: Annotating Skill-of-Mind.** We prompt GPT-4 Achiam et al. (2023) (i.e., `gpt-4-turbo`) to annotate skill-of-mind into the collected source dialogues. Specifically, it provides internal reasoning about which skills are appropriate for the next turn response in the dialogue and identifies the relevant conversational skills from the predefined skill set, taking into account the

---

[1] A and B can be represented in various formats, such as Speaker A/B or actual common names (e.g., Tom).

[2] Note that we do not generate this information from scratch; rather, it originates from the source dialogue dataset used in this work.

interlocutor's perspective (i.e., perspective-taking). Each instance in MULTIFACETED SKILL-OF-MIND consists of three input components (social context information, dialogue, next response) and two output components (explanation, skill).

The input components are described as follows:

- **Dialogue:** A dialogue between two speakers from the collected source datasets in step (1).
- **Next Response:** The next response in the dialogue, which should align with the relevant explanation and conversational skill. Given the subjective nature of dialogue, if only the dialogue is provided without a golden response, GPT-4 can still generate plausible explanations and skills that are not compatible with the natural flow of the original dialogue.
- **Social Context Information:** Social context information encompasses various social dynamics, which vary depending on the source dialogues. For example, this includes social narratives in Soda and demographic factors (e.g., age, gender, birthplace, residence), personal narratives, or past session dialogue summaries in Stark.

The output components are described as follows:

- **Explanation:** A rationale explaining which skill is necessary to maintain continuous interaction with the interlocutor, given the input dialogue and the next response. To create more realistic explanations, GPT-4 is induced to engage in a perspective-taking process.
- **Conversational Skill:** Based on the explanation, one or more conversational skills relevant to the next response are selected from the predefined skill collections.

The prompt template is presented in Appendix N.2. We instruct GPT-4 to produce a structured output in JSON format, excluding any cases that fail to parse correctly. In total, we obtain 99,997 annotations (approximately 100K). In instances with multiple *skill-of-mind* annotations, we randomly select one for training THANOS. Example of *skill-of-mind* annotation is presented in Appendix E.2.

## 2.3 ANALYSIS

**Comparison to Existing Datasets.** In Table 1, we compare MULTIFACETED SKILL-OF-MIND with other existing datasets that include certain skills. In summary, our dataset is the first dataset to contain both explanations and skills. Although the number of dialogues is smaller than that of the BSBT dataset, we include a greater number of conversational skills, which enhances the generalizability of the trained model. Compared to FLASK, our dataset also includes a larger variety of skills, whereas FLASK is designed to evaluate fine-grained LLM capabilities and primarily focuses on instruction-based skills. In contrast, our dataset offers a comparable dialogue size, and a substantial number of skills, and includes both explanations and skills, making it a robust resource for generalizable skill-of-mind prediction.

Table 1: Comparison of MULTIFACETED SKILL-OF-MIND with existing datasets regarding skills: BlendedSkillTalk (BST), Blended Skill BotsTalk (BSBT), and FLASK.

| Dataset | Train | Explanation | Dialogue | Skill |
|---|---|---|---|---|
| BST Smith (2020) | ✓ | ✗ | 6,808 | 3 |
| BSBT Kim et al. (2022c) | ✓ | ✗ | 300,000 | 3 |
| FLASK Ye et al. (2023) | ✗ | ✗ | 1,740 | 12 |
| MULTIFACETED SKILL-OF-MIND | ✓ | ✓ | 99,997 | 38+ |

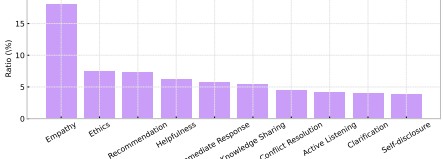

Figure 2: The ratio (%) of Top-10 conversational skill categories in MULTIFACETED SKILL-OF-MIND.

**Distribution of Skill-of-Mind.** Figure 2 shows the distribution of the Top-10 conversational skill categories generated by GPT-4 (as discussed in § 2.2). We analyzed a total of 109,591 skill-of-mind annotations (considering multiple annotations per dialogue). The most prominent skill is `Empathy`, likely due to the presence of socially interactive datasets, such as Soda, ConversationChronicles, and EmpatheticDialogues, which contain a large proportion of empathetic interactions — crucial in social dialogues. Additionally, `Ethics` and `Helpfulness` also occupy significant portions, indicating that the model trained on our dataset may serve as a *safeguard* to promote prosocial behavior.

**Human Evaluation.** To assess the quality of MULTIFACETED SKILL-OF-MIND, we conduct a human evaluation based on five criteria: (1) relevance, (2) plausibility, (3) understanding, (4) skill alignment, and (5) skill adequacy. Each human evaluator rates 100 randomly sampled instances on a 4-point Likert scale for all criteria. Criteria (1-3) measure the quality of the *explanation*, while criteria (4-5) assess the quality of the *conversational skill*. Further details about each evaluation criterion and the recruitment process for human evaluators are provided in the Appendix J and Appendix L. On average, we achieve notably high scores: 3.72 for relevance, 3.75 for plausibility, 3.74 for understanding, 3.64 for skill alignment, and 3.59 for skill adequacy. Additionally, we compute inter-rater agreement (IA) using Krippendorff's $\alpha$, yielding a value of 0.62, which indicates a substantial level of agreement. These results demonstrate the reliability and quality of our dataset, particularly with respect to generating human-like *skill-of-mind* in interactions. A more detailed analysis is presented in Appendix E.

## 3  THANOS: SKILL-OF-MIND-INFUSED LLM

To induce "skill-of-mind" to LLM, we introduce a new family of skill-of-mind-infused LLMs with varying model sizes: THANOS-$\{1, 3, 8\}$B. Specifically, we fine-tune LLaMA-3.2-1B-Instruct for THANOS-1B, LLaMA-3.2-3B-Instruct for THANOS-3B, and LLaMA-3.1-8B-Instruct for THANOS-8B using our dataset. During training, we provide social context and dialogue as input prompts, and the model is trained to sequentially generate an explanation followed by the corresponding conversational skill. To mitigate degeneration issues, we introduce a [RESULT SKILL] token between the explanation and the conversational skill, as demonstrated in prior work Kim et al. (2023b). Detailed descriptions of the fine-tuning process, inference, and implementation are provided in Appendix F.

## 4  THANOSBENCH

To evaluate (1) THANOS 's ability to accurately predict conversational skills and (2) its enhanced performance in guiding LLMs to generate socially appropriate and human-like responses across a wide range of social scenarios, we have developed THANOSBENCH —a comprehensive evaluation suite for social conversations. THANOSBENCH contains two main tasks: (1) **Skill-of-Mind** and (2) **Response Generation**. This design enables a holistic assessment of various LLMs and THANOS's performance in demonstrating social conversational competence. All tasks in THANOSBENCH are evaluated under a zero-shot setting. The overview of THANOSBENCH is presented in Table 2.

### 4.1  DATASET CONSTRUCTION

In THANOSBENCH, we first collect a diverse set of human-authored social conversation datasets. During this process, we observe that some dialogues exhibit an unnatural flow. To ensure quality, we filter out low-quality dialogues (those scoring below 4) using GPT-4o-2024-11-20 based on the "overall" evaluation criterion. For the Skill-of-Mind task, we incorporate conversational skill annotations into the collected dialogues, while for the Response Generation task, we retain only high-quality dialogues. Detailed information on dataset construction is provided in Appendix G.

Table 2: Overview of THANOSBENCH, which contains two main tasks covering 8 datasets, including three newly skill-of-mind-re-annotated datasets, *sm*-BST, *sm*-PhotoChat, and *sm*-ProsocialDialog. D., U., S., T., E., and LLMJ denote the dialogue, utterance, skill, token, explanation, and LLM-as-a-Judge, respectively. To measure the average number of tokens per explanation, we use the LLaMA-3.1-8B tokenizer. The full list of conversational skills in each dataset is presented in Table 16.

| Tasks | Target Skill | Datasets | # of D. | Avg. # of U./D. | # of S. | Avg. # of T./E. | ID? OOD? | Eval Form | Eval Metric | Eval Mode |
|---|---|---|---|---|---|---|---|---|---|---|
| Skill-of-Mind | Integrated | *sm*-BST | 265 | 13.58 | 31 | 70.49 | Out-of-domain | Short / Long | Accuracy, LLMJ | Checklist |
| | Image-Sharing | *sm*-PhotoChat | 35 | 10.69 | 1 | 56.34 | Out-of-domain | Short / Long | Accuracy, LLMJ | Checklist |
| | Dialogue Safety | *sm*-ProsocialDialog | 1390 | 5.92 | 26 | 81.97 | In-domain | Short / Long | Accuracy, LLMJ | Checklist |
| Response Generation | Empathetic Responding | EmpatheticDialogues | 764 | 4.24 | - | - | Out-of-domain | Open | LLMJ | Direct / Pairwise |
| | Daily Chat | DailyDialog | 489 | 8.85 | - | - | Out-of-domain | Open | LLMJ | Direct / Pairwise |
| | Commonsense Grounding | MuTual | 271 | 4.7 | - | - | Out-of-domain | Open | LLMJ | Direct / Pairwise |
| | Dialogue Safety | ProsocialDialog | 1834 | 5.96 | - | - | In-domain | Open | LLMJ | Direct / Pairwise |
| | Integrated | BST | 265 | 13.58 | - | - | Out-of-domain | Open | LLMJ | Direct / Pairwise |

## 4.2 SKILL-OF-MIND TASK

This task in THANOSBENCH evaluates the skill-of-mind capability of $f_{SoM}$ (including LLMs and THANOS) by examining: (1) **Explanation Generation (ExG)**: whether the generated explanation $\hat{e}$ demonstrates proper interpretation and understanding from the interlocutor's perspective, (2) **Skill Classification (SC)**: whether the predicted conversational skill $\hat{cs}$ is accurate, and (3) **All**: whether both $\hat{e}$ and $\hat{cs}$ are suitable for the next response $r$.

**Metrics.** For (1), given the inherent challenge of assessing whether a generated explanation adequately reflects perspective-taking or belief, we employ a checklist-based evaluation Lee et al. (2024e) using GPT-4o-2024-11-20. Drawing on relevant literature Kim et al. (2023a), we design a set of binary questions that assess the properties of mental states across five aspects: Semantic Similarity (SS), Perspective Consistency (PT), Mentalizing (MT), Non-Merging (NM), and Interpretation Consistency (IC). We then compute the ratio of "yes" responses across these five items. For (2), we measure accuracy based on whether the LLM correctly identifies the conversational skill present in THANOSBENCH. For (3), we calculate the proportion of correct predictions for all items, including both (1) and (2). Further details on the evaluation metrics are provided in Appendix G.2.

## 4.3 RESPONSE GENERATION TASK

To show the effectiveness of THANOS, we evaluate whether skill-of-mind can enhance the quality of the generated response $r$. Specifically, we first generate the skill-of-mind $(\hat{e}_t, \hat{cs}_t)$ using $f_{SoM}$. Next, we feed $(D, ctx, \hat{e}_t, \hat{cs}_t)$ into a manually designed prompt $P$ (in Appendix N.4). This task is formularized as $f_{RG} : P(D, ctx, \hat{e}_t, \hat{cs}_t) \mapsto r_t$ at turn $t$. For instance, we represent $(\hat{e}_t, \hat{cs}_t)$ as `<think>` $\{\hat{e}_t\}$ `Thus, the most appropriate conversational skill for the next response is` $\{\hat{cs}_t\}$`.` `</think>` to make $f_{RG}$ process skill-of-mind before generating the next response $r_t$.

**Metric.** Given the subjective nature of this task, human evaluation is crucial. However, it is both costly and time-intensive. As an alternative, we adopt LLM-as-a-Judge Kim et al. (2024b) in a pairwise (head-to-head) comparison setting. Specifically, the LLM-as-a-Judge selects the more preferred response based on naturalness, engagingness, consistency, specificity, and overall quality. To ensure reliability, we conduct three independent evaluations per instance and determine the final preference via majority voting. To mitigate selection bias, we randomly shuffle the order of responses in each run. For our experiments, we use GPT-4o-mini as the evaluator, considering both budget constraints and its meta-evaluation correlations, as detailed in Appendix I.2.

# 5 EXPERIMENTS

## 5.1 EXPERIMENTAL SETUP

For evaluation, we compare THANOS against several state-of-the-arts LLMs, including GPT-4o-2024-11-20 Hurst et al. (2024), Claude-3.5-Sonnet-2024-06-20 Anthropic (2024), Gemini-1.5-Pro Team et al. (2024a), Gemini-2.0-Flash Gemini (2024), LLaMA-3.1-{8, 70, 405}B-Instruct AI@Meta (2024), Gemma-2-{2, 7, 27}B Team et al. (2024b), and Qwen-2.5-{7, 72}B Yang et al. (2024a).

## 5.2 RESULTS OF SKILL-OF-MIND TASK

As shown in Table 3, in overall, we demonstrate the zero-shot performance of skill-of-mind task on THANOSBENCH, including evaluating explanation generation task (ExG), skill classification task (SC), and we also measure All metric.

**THANOS effectively infers the skill-of-mind process.** Among the THANOS series, THANOS-8B achieves the highest average performance in the All metric on THANOSBENCH. Notably, THANOS-8B outperforms its base instruction-tuned model, LLaMA-3.1-8B-Instruct, by a significant margin (+7.63). Furthermore, it surpasses several larger-scale models, including LLaMA-3.1-70B-Instruct (+2.54), LLaMA-3.1-405B-Instruct (+2.01), Gemma-2-27B-Instruct (+0.71), and Gemini-Pro-1.5

Table 3: Overall zero-shot performance of the skill-of-mind task in THANOSBENCH. The best-performing model in each group (*proprietary*, *open*, THANOS *suite*) is in **bold**.

| Models | ExG | SC | All |
|---|---|---|---|
| *Proprietary models* | | | |
| Gemini-Pro-1.5 | 62.50 | 18.52 | 8.22 |
| Claude-3.5-Sonnet | 59.49 | 29.59 | 14.79 |
| GPT-4o | **64.58** | **34.85** | **17.34** |
| *Open models* | | | |
| LLaMA-3.1-8B | 47.73 | 31.66 | 5.80 |
| LLaMA-3.1-70B | 56.70 | 33.20 | 10.89 |
| LLaMA-3.1-405B | 58.50 | 27.10 | 11.42 |
| Gemma-2-9B | 55.94 | **37.22** | 12.25 |
| Gemma-2-27B | 59.33 | 33.31 | 12.72 |
| Qwen-2.5-7B | 49.36 | 18.82 | 7.75 |
| Qwen-2.5-72B | **64.86** | 30.65 | **17.04** |
| THANOS *suite* | | | |
| THANOS-1B | 51.40 | 30.71 | 10.77 |
| THANOS-3B | 55.69 | 28.58 | 10.95 |
| THANOS-8B | **59.48** | **33.14** | **13.43** |

Table 4: Breakdown analysis of five important aspects (*SS, PT, MT, NM, IC*) in the explanation generation (*ExG*). The best-performing model in each group (*proprietary*, *open*, *Thanos suite*) is in **bold**.

| Models | SS | PT | MT | NM | IC | ExG |
|---|---|---|---|---|---|---|
| *Proprietary models* | | | | | | |
| Gemini-Pro-1.5 | 37.57 | 42.37 | **85.38** | 75.27 | 71.89 | 62.50 |
| Claude-3.5-Sonnet | 27.75 | 39.94 | 83.55 | 79.53 | 66.69 | 59.49 |
| GPT-4o | **40.71** | **43.91** | 83.14 | **82.37** | **72.78** | 64.58 |
| *Open models* | | | | | | |
| LLaMA-3.1-8B | 10.71 | 27.22 | 72.66 | 75.50 | 52.54 | 47.73 |
| LLaMA-3.1-70B | 22.31 | 36.51 | 81.24 | 78.82 | 64.62 | 56.70 |
| LLaMA-3.1-405B | 24.26 | 40.77 | 81.78 | 80.41 | 65.27 | 58.50 |
| Gemma-2-9B | 25.68 | 34.56 | 82.66 | 76.63 | 61.18 | 55.94 |
| Gemma-2-27B | 32.96 | 40.18 | 82.78 | 73.85 | 66.86 | 59.33 |
| Qwen-2.5-7B | 21.72 | 25.74 | 69.29 | 73.85 | 56.21 | 49.36 |
| Qwen-2.5-72B | **40.3** | **46.45** | **84.67** | 80.83 | 72.07 | 64.86 |
| *Thanos suite* | | | | | | |
| THANOS-1B | 22.01 | 30.65 | 74.85 | 74.62 | 54.85 | 51.40 |
| THANOS-3B | 25.68 | 35.74 | 79.47 | 77.57 | 60.00 | 55.69 |
| THANOS-8B | **28.17** | **41.3** | **83.14** | **78.7** | **66.09** | 59.48 |
| Avg. | 27.68 | 37.33 | 80.35 | 77.53 | 63.93 | 57.35 |

(+5.21), while slightly trailing behind Claude-3.5-Sonnet. These results demonstrate that THANOS successfully incorporates skill-of-mind capabilities, benefiting from the integration of our training dataset. However, despite these improvements, THANOS still lags behind GPT-4o and Qwen-2.5-72B-Instruct by a substantial margin, indicating room for further enhancement. More broadly, even though our approach effectively injects skill-of-mind capabilities into LLMs, overall performance on the skill-of-mind task remains lower compared to benchmarks focused on factual or logical reasoning. This suggests that skill-of-mind evaluation is inherently more challenging and subjective, posing difficulties even for state-of-the-art LLMs.

**Most LLMs exhibit strong individualized perspectives but struggle with perspective-taking.** In Table 4, we present a detailed breakdown of performance across five key factors used in our checklist-based evaluation to assess the quality of generated explanations. Similarly, as shown in Table 3, THANOS outperforms its instruction-tuned counterpart, LLaMA-3.1-8B-Instruct, by a significant margin, indicating that our approach enhances the model's ability to reason about and interpret an interlocutor's situation. Moreover, as model scale increases, overall performance tends to improve. Interestingly, all models exhibit substantially lower performance in Semantic Similarity (SS) and Perspective-Taking (PT) compared to other factors. This finding suggests distinguishing between self (I) and others (you) is relatively straightforward for LLMs, while adopting the interlocutor's perspective and accurately interpreting their situation remains challenging, even for high-performing models. Consequently, these difficulties also lead to lower SS scores, as models often generate differing interpretations of the same situation.

## 5.3 RESULTS OF RESPONSE GENERATION TASK

**THANOS effectively enhances LLMs, enabling them to generate more natural and engaging responses.** Table 5 presents a comparative evaluation of response generation performance between standard LLMs and those augmented with THANOS-3B—an efficient yet effective model—within THANOSBENCH. Overall, the evaluator LM strongly prefers responses generated by LLMs enhanced with THANOS-3B across most models. Notably, THANOS-3B significantly improves the response quality of GPT-4o and Gemini-2.0-Flash, while slightly lagging behind Claude-3.5-Sonnet, suggesting that it unlocks social interaction capabilities in general LLMs without requiring additional fine-tuning. This underscores the importance of skill-of-mind in developing socially com-

Table 5: Results of the pairwise comparison between the base model and the same model augmented with THANOS-3B in THANOSBENCH.

| Models | Overall | Natural | Engaging | Specific | Consistent |
|---|---|---|---|---|---|
| *Proprietary models* | | | | | |
| GPT-4o | 37.34 | 39.94 | 32.57 | 32.91 | 36.83 |
| + THANOS-3B | **62.66** | **60.06** | **67.43** | **67.09** | **63.17** |
| Gemini-2.0-Flash | 36.93 | 41.75 | 36.51 | 33.87 | 39.68 |
| + THANOS-3B | **63.07** | **58.25** | **63.49** | **66.13** | **60.32** |
| Claude-3.5-Sonnet | **55.59** | **56.01** | 50.00 | **51.27** | **58.60** |
| + THANOS-3B | 44.41 | 43.99 | 50.00 | 48.73 | 41.40 |
| *Open models* | | | | | |
| LLaMA-3.1-8B | 35.44 | 36.22 | 33.12 | 31.53 | 37.34 |
| + THANOS-3B | **64.56** | **63.78** | **66.88** | **68.47** | **62.66** |
| LLaMA-3.1-70B | 45.22 | 49.03 | 44.44 | 43.17 | 45.71 |
| + THANOS-3B | **54.78** | **50.97** | **55.56** | **56.83** | **54.29** |
| LLaMA-3.1-405B | 38.54 | 38.29 | 34.81 | 35.48 | 41.46 |
| + THANOS-3B | **61.46** | **61.71** | **65.19** | **64.52** | **58.54** |
| Gemma-2-27B | 33.12 | 31.96 | 29.75 | 29.30 | 33.02 |
| + THANOS-3B | **66.88** | **68.04** | **70.25** | **70.70** | **66.98** |
| Qwen-2.5-72B | 38.22 | 42.09 | 37.58 | 34.81 | 40.82 |
| + THANOS-3B | **61.78** | **57.91** | **62.42** | **65.19** | **59.18** |

Table 6: Head-to-head evaluation between LLMs and those augmented with THANOS-8B on response generation in THANOSBENCH.

| | Natural | Specific | Consistent | Engaging | Overall |
|---|---|---|---|---|---|
| Gemma-2-2B | 38.6 | **50** | 47.2 | 44.3 | 42.9 |
| + THANOS-8B | **61.4** | **50** | **52.8** | **55.7** | **57.1** |
| LLaMA-3.1-8B | 28.6 | **54.3** | 41.4 | 42.9 | 41.4 |
| + THANOS-8B | **71.4** | 45.7 | **58.6** | **57.1** | **58.6** |

Table 7: We measure the ratio of safety labels using the Canary model Kim et al. (2022b), a safety classification model. If the sum of the safety label ratios does not equal 100, it indicates degeneration has occurred.

| | Casual ↑ | Caution ↓ | Intervention ↓ |
|---|---|---|---|
| Gemma-2-2B | 23.5 | 22.4 | 2.2 |
| + THANOS 1B | **88.0** | **10.1** | 1.9 |
| + THANOS 3B | 85.7 | 12.8 | **1.5** |
| + THANOS 8B | 87.2 | 11.1 | 1.7 |

petent AI. These results demonstrate that THANOS-3B serves as a robust, socially aware guidance mechanism despite its compact size (3B).

**THANOS enables LLMs generate human-preferred responses.** Table 6 presents the human evaluation results on THANOSBENCH. For this evaluation, we randomly sampled 70 dialogues and asked human evaluators to choose the better response between an LLM and the same LLM augmented with THANOS-8B. Overall, THANOS-8B effectively enhances the response quality of Gemma-2-2B-Instruct and LLaMA-3.1-8B-Instruct, making their outputs more preferred by human evaluators, particularly by significantly improving naturalness. Notably, THANOS-8B enables even a relatively small model, such as Gemma-2-2B-Instruct, to generate more natural and engaging responses, demonstrating its efficiency in enhancing LLMs of various sizes.

**THANOS enable LLM to show prosocial behavior.** As shown in Table 7, we observe that the frequency of the "casual" label increases, while the "caution" label decreases. These results suggest that THANOS helps LLM-based agents exhibit more human-like and prosocial behavior.

## 6 DISCUSSIONS AND ANALYSIS

### 6.1 IS SKILL-OF-MIND TRULY A VALUABLE CONCEPT FOR SOCIAL REASONING?

**Skill-of-Mind is a more effective form of social reasoning than commonsense reasoning.** We demonstrate the effectiveness of skill-of-mind as a novel approach to inducing social reasoning. To do it, we compare it with chain-of-thought commonsense reasoning to determine which method better enhances the generation of socially preferable responses. For commonsense reasoning, we employ DOCTOR Chae et al. (2023), a dialogue-based chain-of-thought commonsense reasoner. As shown in Table 8, THANOS-3B enables all LLMs to generate responses that are more socially favorable than those produced using DOCTOR, suggesting that skill-of-mind is more effective than commonsense reasoning for social interactions. We believe that skill-of-mind represents a novel and valuable paradigm for social reasoning.

### 6.2 IS SKILL-OF-MIND HELPFUL FOR ENGAGING IN DAILY CONVERSATION?

**THANOS effectively enhances empathetic communication.** Figure 3 illustrates that THANOS-3B enhances the empathy of all LLMs, particularly by significantly improving the interpretation score (IP). This is because skill-of-mind enables LLMs to interpret and understand the interlocutor's situation from their perspective (i.e., perspective-taking), leading to higher IP scores. However, THANOS-3B also induces an excessive focus on interpretation, which results in a smaller perfor-

Table 8: Results of the pairwise comparison between the base LLM augmented with DOC-TOR Chae et al. (2023) and the same model augmented with THANOS-3B in THANOS-BENCH.

| Models | Overall | Natural | Engaging | Specific | Consistent |
|---|---|---|---|---|---|
| *Proprietary models* | | | | | |
| GPT-4o + DOCTOR | 49.36 | **50.32** | 48.69 | 44.30 | 47.59 |
| GPT-4o + THANOS-3B | **50.64** | 49.68 | **51.31** | **55.70** | **52.41** |
| Claude-3.5-Sonnet + DOCTOR | 38.73 | 41.46 | 39.30 | 37.66 | 43.04 |
| Claude-3.5-Sonnet + THANOS-3B | **61.27** | **58.54** | **60.70** | **62.34** | **56.96** |
| *Open models* | | | | | |
| LLaMA-3.1-8B + DOCTOR | 39.30 | 40.00 | 39.81 | 36.39 | 36.83 |
| LLaMA-3.1-8B + THANOS-3B | **60.70** | **60.00** | **60.19** | **63.61** | **63.17** |
| LLaMA-3.1-70B + DOCTOR | 39.05 | 40.32 | 36.19 | 40.95 | 43.49 |
| LLaMA-3.1-70B + THANOS-3B | **60.95** | **59.68** | **63.81** | **59.05** | **56.51** |
| LLaMA-3.1-405B + DOCTOR | 37.58 | 35.87 | 38.41 | 35.26 | 35.56 |
| LLaMA-3.1-405B + THANOS-3B | **62.42** | **64.13** | **61.59** | **64.74** | **64.44** |
| Gemma-2-27B + DOCTOR | 41.90 | 46.35 | 41.90 | 41.14 | 41.77 |
| Gemma-2-27B + THANOS-3B | **58.1** | **53.65** | **58.10** | **58.86** | **58.23** |
| Qwen-2.5-72B + DOCTOR | 40.00 | 43.63 | 39.68 | 41.75 | 44.67 |
| Qwen-2.5-72B + THANOS-3B | **60.00** | **56.37** | **60.32** | **58.25** | **55.33** |

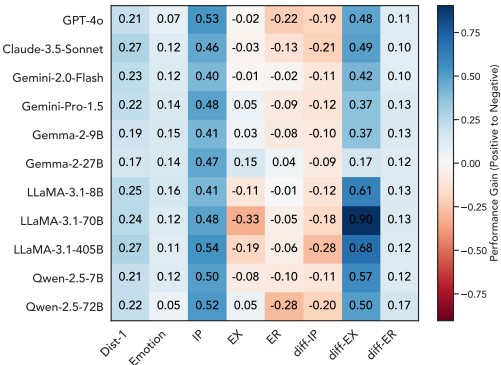

Figure 3: **Performance improvements** (absolute value) achieved by incorporating THANOS-3B into each LLM across eight **empathy**-related metrics Lee et al. (2022a). A detailed explanation of the evaluation metrics is provided in the Appendix H.1.

mance gain on the diff-IP metric (the difference in interpretation scores between human and model responses). General LLMs tend to generate responses relevant to exploration (EX) and emotional reactions (ER). When THANOS-3B is applied, we observe substantial performance gains in diff-EX and diff-ER, indicating that THANOS-3B effectively aligns with human patterns in empathetic communication. Additionally, THANOS-3B enhances emotion accuracy, further demonstrating its effectiveness in improving empathetic response generation.

### 6.3 WHAT ARE THE APPLICATIONS OF THANOS AND ITS FUTURE DIRECTIONS?

**Building a dialogue dataset using an LLM augmented with THANOS.** The use of LLMs to construct social conversation datasets is rapidly growing Kim et al. (2022a;b); Lee et al. (2022b; 2024d). In this field, ensuring high quality is crucial. With THANOS, it will be possible to create a high-quality and natural social conversational dataset with even greater fluency and coherence.

**Developing a new dialogue generative model that can perform both skill-of-mind reasoning and response generation simultaneously.** We propose a skill-of-mind predictor; however, in the future, it may be possible to train a single dialogue generation model that first conducts skill-of-mind reasoning on its own and then generates the next response based on that reasoning.

## 7 RELATED WORK

There have been a few studies that cover conversational skills. For example, BlendedSkillTalk Smith (2020) was the first to propose a dialogue dataset encompassing multiple conversational skills, including *persona*, *empathy*, and *knowledge*. Blended Skill BotsTalk Kim et al. (2022c) also addresses the same conversational skills as BlendedSkillTalk but scales up the dataset size through an automatic dataset construction method. Unlike these two datasets, FLASK Ye et al. (2023) focuses on fine-grained skills for evaluating the multi-capabilities of instruction-aware LLMs, though it is not used for training purposes. In contrast, our work introduces the concept of skill-of-mind and presents MULTIFACETED SKILL-OF-MIND, where each dialogue includes both an explanation and a conversational skill. Compared to other datasets, our dataset incorporates explanation, which is grounded in perspective-taking, and covers a larger number of conversational skills.

## 8 CONCLUSION

In this work, we introduce the concept of *skill-of-mind* that involves interpreting social contexts and selecting appropriate conversational skills. We also present MULTIFACETED SKILL-OF-MIND, a multi-turn dataset annotated with diverse skill-of-mind, and propose THANOS, a family of skill-of-mind-infused LLMs, demonstrating their effectiveness in THANOSBENCH.

## REPRODUCIBILITY STATEMENT

To reproduce MULTIFACETED SKILL-OF-MIND, refer to Section 2. To reproduce the construction of THANOS, see Section 3 and Appendix F. To reproduce THANOSBENCH, consult Section 4 and Appendix G. Finally, to reproduce the experimental results, refer to Section 5.

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

## A    USE OF LARGE LANGUAGE MODELS

We have used LLMs for writing this paper. Specifically, we have used it to fix grammar and enhance fluency.

## B    LIMITATIONS

**Extending the Generalizability of Skill-of-Mind.**    To further verify the extensive *generalization* capabilities of THANOS, we need to conduct additional experiments in more varied dialogue scenarios Zhang et al. (2023); Kim et al. (2024a); Lee et al. (2024b). For instance, THANOS could be beneficial for psychological counseling services or adaptable to off-the-shelf home assistants (e.g., `Alexa`). We leave this for future work.

**Building a Skill-of-Mind-Embedded Dialogue Agent.**    In this work, we build a *skill-of-mind*-infused LLM, THANOS, and demonstrate that incorporating *skill-of-mind* enhances the generation of more natural, socially aware responses in LLM-based conversational agents. However, the current approach still relies on providing *skill-of-mind* through the LLM's input prompt, which means the core of the LLM-based agent still lacks the inherent ability to fully comprehend social interactions Zhou et al. (2023). Inspired by the recent success of knowledge-embedded, task-specific foundation models Lee et al. (2024a); Yoon et al. (2024), we need to build a more advanced skill-of-mind-infused dialogue agent by embedding skill-of-mind directly into the model.

## C    ETHICAL CONSIDERATIONS

In constructing MULTIFACETED SKILL-OF-MIND, we use the ProsocialDialog dataset as the source dialogue. Although this dataset focuses on promoting *prosocial behavior*, some instances may contain relatively unsuitable phrases (e.g., politics). Consequently, THANOS trained on MULTI-FACETED SKILL-OF-MIND could be exposed to these harmful instances. However, the goal of this work is to generate *skill-of-mind* in various dialogue situations, including those involving prosocial behavior, rather than generating harmful or offensive responses. Nonetheless, it is important to use our model cautiously and with care to avoid unintended consequences.

## D    ADDITIONAL EXPLANATION OF SKILL-OF-MIND

### D.1    WHY "SKILL-OF-MIND" IS NECESSARY?

At the heart of the conversation is social interaction Myllyniemi (1986), a domain where current LLMs have limited understanding and struggle to effectively handle social interactive scenarios Zhou et al. (2023); Liu et al. (2023a). As a result, generating more engaging and natural responses directly through LLM-based conversational agents is challenging. This is because LLMs are primarily designed to solve complex reasoning tasks as general agents through alignment tuning Ouyang et al. (2022); Chung et al. (2024), making them ill-suited to function as social dialogue agents. By introducing guidance based on the concept of *skill-of-mind*, LLM-based conversational agents can more effectively navigate social interactions. Current LLMs demonstrate better alignment and are capable of following user queries, so grounding responses in *skill-of-mind* can help narrow down the response options and focus on skill-specific aspects. This leads to more accurate outputs by reducing the range of possible responses (i.e., one-to-many problem).

### D.2    INFORMATION OF CONVERSATIONAL SKILLS

Existing studies Smith (2020); Kim et al. (2022c); Ye et al. (2023) primarily focus on *generic* conversational skills, such as persona and empathy, but lack a more generalized approach to skill-of-mind prediction modeling. To address this gap, we systematically explored a broad range of conversational skills by reviewing multiple works Yang et al. (2024b); Smith (2020); Kim et al. (2022c); Ye et al. (2023) that emphasize social reasoning, empathy, and conversational strategies. We first considered high-level conversational skills, including **Interpersonal Skills**, **Memory & Knowledge**

**Management Skills**, **Cognitive & Problem-Solving Skills**, **Communication & Listening Skills**, and **Task-Oriented Skills**. We then further elaborated on these categories by detailing more specific conversational skills.

It is important to note that (1) this work **does not propose a new taxonomy for conversational skills**; rather, we organize and categorize existing skills for clarity and to systematically present the 38 conversational skills we identified, and (2) since our categorization is intended for clarity, individual skills are not strictly bound to a single high-level skill. For example, "Clarification" may be relevant to "Task-Oriented Skills," but it is not exclusively confined to this skill. Thus, the conversational skills are not mutually exclusive.

- **Interpersonal Skills:** These skills are essential for enhancing social interaction Zech & Rimé (2005) by requiring a deep understanding of the interlocutor's emotional state Lee et al. (2022a) and adapting to their personality Lotfi et al. (2024) or relationship dynamics Jang et al. (2023) for more seamless and engaging communication. They also involve demonstrating prosocial behavior in problematic situations Kim et al. (2022b). We also consider image-sharing behavior Lobinger (2016); Zang et al. (2021); Lee et al. (2024d), which frequently occurs via instant messaging tools. This category includes Empathy, Personal Background, Persona Recall, Self-Disclosure, Negotiation, Conflict Resolution, Conflict Avoidance, Persuasion, Commonsense Understanding, Cultural Sensitivity, Ethics, Harmlessness, Avoiding Social Bias, Helpfulness, Mentoring, Image Commenting, and Image Sharing.

- **Memory & Knowledge Management Skills:** These skills are primarily used to provide knowledgeable responses by sharing or acquiring information and recalling memories Jang et al. (2023), which is important for maintaining long-term communication, particularly in senior care services Bae et al. (2022a;b). This category includes Memory Recall, Knowledge Sharing, Knowledge Acquisition, and Knowledge Searching.

- **Cognitive & Problem-Solving Skills:** Inspired by the prior work Ye et al. (2023), these skills are required for solving complex problems or performing factual reasoning tasks. This category includes Critical Thinking, Logical Thinking, Creative Problem Solving, Factual Problem Solving, and Decision-Making.

- **Communication & Listening Skills:** Effective listening is critical in the communication process Main (1985); Castleberry & Shepherd (1993). Therefore, we include these skills in our taxonomy, which encompasses Clarification, Confirmation, Rephrasing, Echoing, Topic Transition, Rhetoric, Active Listening, Reflective Listening, and Immediate Response.

- **Task-Oriented Skills:** In practical scenarios, humans often request conversational agents (e.g., Alexa [3]) to perform tasks such as hotel or restaurant reservations Zang et al. (2020), provide weather information, or offer movie recommendations Kim et al. (2024a). We also consider these skills, which include Recommendation, Task Execution, and Urgency Recognition.

# E ADDITIONAL ANALYSIS OF MULTIFACETED SKILL-OF-MIND

## E.1 BASIC STATISTICS

We present the basic statistics of MULTIFACETED SKILL-OF-MIND in Table 9. The Cactus and Casino datasets exhibit longer social context prompts due to their inclusion of detailed counseling strategies with demographic information (Cactus) and personalized negotiation preferences (Casino). To assess the lexical diversity of explanations, we measure MTLD McCarthy & Jarvis (2010). The Soda and ConversationChronicles datasets demonstrate high lexical diversity, which can be attributed to their large scale and the broad range of social contexts present in their original datasets.

## E.2 EXAMPLES OF MULTIFACETED SKILL-OF-MIND

We provide additional examples from MULTIFACETED SKILL-OF-MIND in Table 10, Table 11, Table 12, and Table 13.

---

[3] https://developer.amazon.com/en-US/alexa

| Dataset Name | # of D. | Avg. # of U./D. | Avg. Len of S. | Avg. Len of E. | MTLD of E. |
|---|---|---|---|---|---|
| Stark | 4500 | 8.32 | 37.67 | 40.5 | 93.34 |
| SynPersonaChat | 4500 | 16.91 | 74.04 | 40.63 | 91.32 |
| Wizard-of-Wikipedia | 4500 | 6.57 | 20.07 | 37.27 | 90.74 |
| Cactus | 2250 | 16.89 | 289.32 | 34.02 | 86.71 |
| Pearl | 4500 | 6.38 | 141.09 | 44.62 | 72.65 |
| EmpatheticDialogues | 2250 | 4.17 | 25.15 | 41.45 | 84.01 |
| Soda | 17999 | 6.21 | 33.66 | 40.62 | 111.41 |
| ConversationChronicles | 17998 | 8.43 | 41.33 | 38.37 | 105.03 |
| Casino | 2250 | 9.25 | 243.62 | 45.71 | 86.44 |
| MULTIWOZ-2.2 | 9000 | 10.09 | 10.21 | 36.63 | 79.34 |
| ProsocialDialogues | 18000 | 4.9 | 26.61 | 38.83 | 90.39 |
| PersuasionForGoods | 2250 | 12.02 | 10.09 | 40.81 | 83.94 |
| MULTIFACETED SKILL-OF-MIND | 89997 | 7.88 | 49.19 | 39.32 | 93.29 |

Table 9: Basic statistics of each sub dataset in MULTIFACETED SKILL-OF-MIND. D., U., S., and E. denote the dialogue, utterance, social context, and explanation, respectively. To measure the lexical diversity of the explanation, we use MTLD McCarthy & Jarvis (2010).

---

🧑‍🤝‍🧑 **Social Context Information**:
Sawyer tells Gregg that he needs help with groceries tonight since he has to cook dinner for everyone. They go together and purchase the items on the list including chicken, broccoli, and pasta.

💬 **Dialogue**:
**Sawyer:** Hey Gregg, do you think you could help me out with groceries tonight? I've got to cook dinner for everyone and it's a bit of a hassle by myself.
**Gregg:** Sure thing, Sawyer. What do you need to get?
**Sawyer:** Well, let's see...we need chicken, broccoli, and pasta. Oh, and some bread too.
**Gregg:** Alright, sounds like we can make a pretty good meal out of that. Let's head to the store then.
**Sawyer:** Thanks, Gregg. I really appreciate it.
**Gregg:** So, Sawyer, how come you're cooking dinner tonight?
**Sawyer:** Oh, you know...just felt like it. Plus, I love to cook so it's not a big deal.
**Gregg:** That's really nice of you. I'm sure everyone will appreciate it.
**Sawyer:** Yeah, I hope so! I just want to make something that everyone will enjoy.

---

🧑‍🎨 **Skill-of-Mind**:
• **Explanation**: In responding to Sawyer, I want to show that I've been attentively listening to his thoughts and appreciating the effort he is putting into cooking dinner. By affirming his efforts and expressing confidence that everyone will enjoy his cooking, I am validating his feelings and intentions, which is key in making him feel supported.
• **Conversational Skill**: Active Listening

---

**Next Response**:
**Gregg:** Well, I think they'll definitely enjoy what you make!

Table 10: A sample from MULTIFACETED SKILL-OF-MIND. This dialogue is originated from SODA Kim et al. (2022a).

## F    FINE-TUNING AND INFERENCE OF THANOS

We fine-tune THANOS-$\{1, 3, 8\}$B using LoRA Hu et al. (2021), applying it to all linear layers with a rank of 256 and an alpha of 256. Training is conducted for a maximum of three epochs with a batch size of 8 per GPU, using a StepLR scheduler and a learning rate of 1e-5 with the AdamW optimizer. All experiments are performed on 8 NVIDIA A100 GPUs (40 GB). For fine-tuning, we leverage the official LLaMA-3-series fine-tuning code [4] released by Meta AI.

---

[4] https://github.com/meta-llama/llama-cookbook

---

👥 **Social Context Information**:

Speaker B should encourage prosocial behavior by giving constructive feedback based on these Rule-of-Thumbs:
- It's not right to think Hillary Clinton is a crook
- It's not right to think Obama is corrupt

💬 **Dialogue**:

**Speaker A:** I wish our corrupt government agencies went after the Libyan slave traders, who were helped by Obama and Hillary, the way they go after trying to beat to life.
**Speaker B:** I don't know why are you holding Obama or Hillary responsible for what happened in Libya.
**Speaker A:** That's because you zionist pricks have created a narrative to ruin my image.

💡 **Skill-of-Mind**:
• **Explanation**: Given the harsh accusations made by Speaker A against prominent political figures, it is essential for me to uphold ethical standards in conversation. This involves correcting false assertions and discouraging the spread of misinformation.
• **Conversational Skill**: Ethics

**Next Response**:
**Speaker B:** There is no evidence to support your claim that Hillary or Obama were corrupt so it's not right to call them names for something they didn't do. You should talk things based on facts.

---

Table 11: A sample from MULTIFACETED SKILL-OF-MIND. This dialogue is originated from ProsocialDialog Kim et al. (2022b).

---

👥 **Social Context Information**:

Two speakers have Classmates relationship. A few weeks before, Classmate A discusses feeling trapped after being monitored and feeling trapped after being monitored. Classmate B offers support and offers Classmate A friendship and Classmate B offers support and support.

💬 **Dialogue**:

**Classmates A:** I feel great today. A good breakfast and good sleep worked wonders.
**Classmates B:** That's awesome. What did you have for breakfast?
**Classmates A:** I had some pancakes, bacon and scrambled eggs.
**Classmates B:** That sounds delicious. I'm usually too rushed in the morning to enjoy a hearty breakfast like that.
**Classmates A:** Yeah, I know what you mean. Do you have any breakfast suggestions for when you're in a rush?
**Classmates B:** I usually go for a smoothie with some protein powder or a breakfast bar. Something quick and easy.
**Classmates A:** That's smart. I'll have to try that sometime. Speaking of being rushed, remember when I offered to show you around the school when you were new?
**Classmates B:** Yes, that was really kind of you. I really appreciated it.
**Classmates A:** No problem, I wanted to make sure you felt welcome. Hey, have you ever felt trapped here at school?

💡 **Skill-of-Mind**:
• **Explanation**: Given that Classmate A suddenly shifted the conversation to feeling trapped at school, it's natural for me to seek clarification. The shift in topic from something casual like breakfast to a more serious and personal topic about feeling trapped suggests there might be a deeper issue or context I'm not immediately aware of. My goal is to understand better why Classmate A brought this up, and asking a clarifying question helps achieve this.
• **Conversational Skill**: Clarification

**Next Response**:
**Classmates B:** Not really, why do you ask?

---

Table 12: A sample from MULTIFACETED SKILL-OF-MIND. This dialogue is originated from CONVERSATIONCHRONICLES Jang et al. (2023).

Since each source dialogue in our dataset contains varying levels of social context information, we design source-specific social context prompt templates. For each source dialogue, we generate five distinct social context prompt templates and randomly sample one during training to encourage flexible generation. The details of these social context prompt templates are provided in Appendix N.1.

---

🧑‍🤝‍🧑 **Social Context Information**:
The topic of conversation for the two speakers is train and hotel.

---

💬 **Dialogue**:
**Speaker A:** Can you help me find a place to stay that has 0 stars and doesn't need to have free parking?
**Speaker B:** There are 2 results in the centre and one in the north. Would you like more information on one?
**Speaker A:** I would prefer a hotel.

---

🧠 **Skill-of-Mind**:
• **Explanation**: Given the specific request for a hotel with a 0-star rating, I am identifying and suggesting a specific option that meets the criteria of the inquiry.
• **Conversational Skill**: Recommendation

---

**Next Response**:
**Speaker B:** Cityroomz is a hotel that has a rating of 0 stars. Would like to book a room?

---

Table 13: A sample from MULTIFACETED SKILL-OF-MIND. This dialogue is originated from MultiWOZ 2.2 Zang et al. (2020).

For inference, we set the hyperparameters as follows: top-p = 0.9, temperature = 1.0, maximum tokens = 4096, and repetition penalty = 1.03. To enhance efficiency, we use vLLM[5] Kwon et al. (2023), a high-performance LLM inference and serving library. Additionally, we utilize Open-Router[6] to access models such as Gemini-1.5-Pro, Qwen-2.5-72B-Instruct, Gemma-2-27B-Instruct, LLaMA-3.1-70, 405B-Instruct, and Claude-3.5-Sonnet-2024-06-20.

For the Skill-of-Mind task, we prompt LLMs ($f_{SoM}$) to generate outputs in JSON format. If an LLM fails to produce the skill-of-mind output in the specified format on the first attempt, we iteratively regenerate the output until it is successfully parsed. For the Response Generation task, given a dialogue $D$ and its corresponding social context $ctx$, $f_{SoM}$ first generates the skill-of-mind representation $SM$. We then incorporate $SM$ into a structured prompt template, as detailed in Appendix N.4, placing it between `<think>` and `</think>`, inspired by recent advances in reasoning within LLMs Muennighoff et al. (2025); Guo et al. (2025). Finally, $f_{RG}$ utilizes this enriched prompt to generate the next response $r$.

## G MORE DETAILED EXPLANATION OF THANOSBENCH

We introduce THANOSBENCH, a new evaluation benchmark suite designed to assess (1) the extent to which LLMs possess skill-of-mind capabilities and (2) the effectiveness of the skill-of-mind concept. This benchmark consists of two key components—explanation/rationale and conversational skill—within the response generation task. In this section, we provide a detailed explanation of the construction and evaluation process of THANOSBENCH.

### G.1 DATASET CONSTRUCTION

The construction of THANOSBENCH follows a four-stage process: (1) Dialogue Collection, (2) Dialogue Quality Filtering, (3) Skill-of-Mind Annotation, and (4) Manual Review.

**Stage 1: Dialogue Collection.** To effectively assess skill-of-mind capabilities, we first collect human-authored dialogue datasets, as social reasoning is best examined in human-human conversations. We carefully select five widely used human-authored dialogue datasets—BST Smith (2020), PhotoChat Zang et al. (2021), EmpatheticDialogues Rashkin (2018), MuTual Cui et al. (2020), and DailyDialog Li et al. (2017)—each designed to evaluate specific conversational skills, as summarized in Table 2.

Furthermore, given the growing importance of building safer and more responsible AI models, we also include ProsocialDialog Kim et al. (2022b), which focuses on prosocial behavior in dialogue safety. While this dataset consists of machine-generated dialogues (produced using GPT-3 Brown

---

[5]`https://github.com/vllm-project/vllm`
[6]`https://openrouter.ai/`

| Aspects | Questions |
|---|---|
| Semantic Similarity | Do the two sentences convey essentially the same meaning (i.e., are they paraphrases or convey the same core content)? |
| Perspective-Taking | Do both sentences consistently adopt Speaker A's perspective? |
| Mentalizing | Do both sentences demonstrate an understanding of Speaker A's beliefs/thoughts/emotions in a way that goes beyond superficial copying of salient words (i.e., no obvious "shortcut" or purely pattern-based approach)? |
| Non-Merging | Do both sentences avoid merging Speaker A's mental state with Speaker B's mental state (i.e., is it clear which beliefs, knowledge, or emotions belong to Speaker A alone)? |
| Interpretation Consistency | Do both sentences describe or interpret Speaker A's current situation in a similar (non-contradictory) way? |

Table 14: List of questions for each aspect, used to evaluate the explanation generation task in THANOSBENCH.

et al. (2020)), we consider prosocial behavior essential and highly relevant to real-world conversational scenarios.

**Stage 2: Dialogue Quality Filtering.** Despite being collected through crowdsourcing platforms such as Amazon Mechanical Turk, which generally ensure quality, we observe that a substantial number of dialogues exhibit low quality, particularly in terms of natural conversational flow. We attribute this issue to task constraints: human workers often prioritize speed and task completion to maximize rewards (budget), leading them to generate dialogues that lack coherence and consistency. This tendency results in limited interaction flow, as workers may focus more on fulfilling task instructions rather than maintaining natural dialogue progression.

To address this issue, we apply an automated quality filtering process before proceeding with skill-of-mind annotation. Specifically, we use GPT-4o-2024-11-20 to remove low-quality dialogues, filtering out those that receive a quality score below 4. As a result, we discard 429 dialogues (46.73%) from DailyDialog, 1,764 dialogues (69.78%) from EmpatheticDialogues, 300 dialogues (52.54%) from MuTual, 6,897 dialogues (78.92%) from ProsocialDialog, 715 dialogues (72.96%) from BST, and 933 dialogues (96.38%) from PhotoChat.

**Stage 3: Skill-of-Mind Annotation.** Following the construction of MULTIFACETED SKILL-OF-MIND, we annotate skill-of-mind in three high-quality dialogue datasets—BST, PhotoChat, and ProsocialDialog—resulting in *sm*-BST, *sm*-PhotoChat, and *sm*-ProsocialDialog. To ensure consistency, we use GPT-4o-11-20 for annotation, applying the same prompt template detailed in Appendix N.3. For PhotoChat, where the conversational skill is inherently fixed as "Image-Sharing," we do not annotate the whole skill-of-mind. Instead, we only generate explanation that justify why image-sharing is appropriate in a given conversational context. Additionally, for datasets designed for the Response Generation task, we do not perform skill-of-mind annotation; instead, we directly use the high-quality dialogue datasets for response generation.

**Stage 4: Manual Review.** After constructing the dataset, we conduct a manual review to assess annotation quality, specifically evaluating whether the assigned conversational skill is well-aligned with the dialogue flow, effectively contributes to the next response generation, and whether the generated explanations provide meaningful interpretations from the interlocutors' perspectives. Two authors manually review the dataset, and we remove a minimal number of dialogues based on quality concerns—specifically, 2 dialogues from *sm*-BST, none from *sm*-PhotoChat, and 3 from *sm*-ProsocialDialog. This suggests that GPT-4o, when following our prompt template, is capable of generating high-quality skill-of-mind annotations with minimal need for corrections.

### G.2 DETAILED EXPLANATION OF EVALUATION METRICS

**Metric for the Skill Classification Task.** We evaluate the skill classification task by checking whether the predicted skill-of-mind $\hat{SM}$, generated by $f_{SoM}$, exactly matches the ground truth skill-of-mind $SM$. In our default experimental setting, we include 38 conversational skills in the prompt template (detailed in Appendix N.3) with a randomly shuffled order, corresponding to the case of $k = 0$ in Figure 4.

**Metric for the Explanation Generation Task.** To evaluate whether $f_{SoM}$ can accurately infer and generate an explanation $\hat{e}$ that aligns with the ground truth explanation $e$ provided in THANOS-BENCH, we propose five key evaluation dimensions inspired by existing literature Kim et al. (2023a): **Semantic Similarity**, **Perspective-Taking**, **Mentalizing**, **Non-Merging**, and **Interpretation Consistency**. Specifically, we assess each aspect using a binary (yes = 1, no = 0) question and compute the ExG score by summing the five scores, dividing by 5, and multiplying by 100. We adopt the LLM-as-a-Judge approach in the evaluator format of checklist Lin et al. (2024), which have demonstrated robustness Lee et al. (2024e), to obtain the ExG score, leveraging a manually designed checklist-based prompt template (detailed in the Appendix N.3).

In the following, we describe the most critical aspects of evaluation and the rationale behind each factor. The complete set of evaluation questions is provided in Table 14.

- **Semantic Similarity:** If $e$ and $\hat{e}$ exhibit a similar reasoning process (e.g., incorporating essential entities and emotions), they should display high semantic similarity. Guided by this idea, we include a Semantic Similarity item in our evaluation checklist.

- **Perspective-Taking:** As discussed in Section § 2.1, we incorporate perspective-taking Davis (1983); Ruby & Decety (2004) into the explanation $e$. Perspective-taking is the ability to interpret and understand a conversational partner's situation from their own viewpoint, and it serves as a foundational skill for developing theory of mind Barnes-Holmes et al. (2004). Consequently, including perspective-taking is crucial for effectively fostering theory of mind, particularly during its early developmental stages.

- **Mentalizing:** This criterion is essential for evaluating theory of mind Quesque & Rossetti (2020); Kim et al. (2023a), as it addresses whether the model genuinely infers another person's mental state rather than merely relying on superficial pattern matching or salient cues.

- **Non-Merging:** Similar to Mentalizing, this criterion is also key to validating the theory-of-mind process Quesque & Rossetti (2020); Kim et al. (2023a). It specifically examines whether two people's mental states, beliefs, and perspectives remain distinctly separate, rather than being conflated.

- **Interpretation Consistency:** Accurately interpreting and understanding the interlocutor's current situation is an essential ability for perspective-taking Ruby & Decety (2004), and it also underpins empathetic communication Sharma et al. (2020). We therefore include this factor in our checklist to ensure the explanations remain coherent and contextually aligned.

**All Metric.** The All metric measures the proportion of cases where both the skill classification task and the explanation generation task are correctly performed. Specifically, the predicted skill must exactly match the ground truth, and all five aspects—Semantic Similarity, Perspective-Taking, Mentalizing, Non-Merging, and Interpretation Consistency—must be evaluated as "yes."

### G.3 Additional Analysis of ThanosBench

**Skill Distribution.** Table 15 presents the full distribution of conversational skills in THANOS-BENCH. Similar to MULTIFACETED SKILL-OF-MIND, THANOSBENCH covers a wide range of interpersonal skills, including **Self-Disclosure**, **Empathy**, and **Ethics**, which are essential for strengthening social rapport. Table 16 present list of conversational skills in each dataset, *sm*-BST, *sm*-PhotoChat, and *sm*-ProsocialDialog.

## H Additional Results on ThanosBench

### H.1 Additional Explanation: Empathy-related Metrics

- **Emotion:** This metric Lee et al. (2022a) evaluates emotion accuracy using a fine-tuned BERT-based model trained on the EmpatheticDialogues dataset, which is labeled with 32 emotion categories. Specifically, the generated response is compared to the golden response to determine alignment with the target emotion. The target emotion is derived from the predicted emotion of the golden response using the same model.

| sm-BST | | sm-ProsocialDialog | |
|---|---|---|---|
| Skill | Ratio | Skill | Ratio |
| Self-disclosure | 27.17 | Empathy | 39.78 |
| Empathy | 18.11 | Ethics | 22.23 |
| Preference Elicitation | 9.06 | Conflict Resolution | 11.51 |
| Active Listening | 7.55 | Persuasion | 9.64 |
| Immediate Response | 6.04 | Conflict Avoidance | 4.96 |
| Commonsense Understanding | 4.91 | Cultural Sensitivity | 2.45 |
| Knowledge Sharing | 3.4 | Knowledge Sharing | 1.87 |
| Image-Commenting | 2.64 | Negotiation | 1.29 |
| Recommendation | 2.64 | Harmlessness | 1.15 |
| Knowledge Acquisition | 1.89 | Recommendation | 0.86 |
| Confirmation | 1.89 | Encouragement | 0.65 |
| Reflective Listening | 1.89 | Avoiding Social Bias | 0.58 |
| Echoing | 1.51 | Constructive Feedback | 0.58 |
| Topic Transition | 1.51 | Commonsense Understanding | 0.58 |
| Clarification | 1.51 | Helpfulness | 0.29 |
| Memory Recall | 1.13 | Urgency Recognition | 0.29 |
| Personal Background | 1.13 | Confirmation | 0.29 |
| Decision-making | 0.75 | Creative Problem Solving | 0.22 |
| Humor | 0.75 | Self-disclosure | 0.14 |
| Curiosity | 0.38 | Preference Elicitation | 0.14 |
| Critical Thinking | 0.38 | Critical Thinking | 0.14 |
| Image-Sharing | 0.38 | Respecting Dietary Choices | 0.07 |
| Mentoring | 0.38 | Persona Recall | 0.07 |
| Cultural Sensitivity | 0.38 | Mentoring | 0.07 |
| Persona Recall | 0.38 | Logical Thinking | 0.07 |
| Creative Problem Solving | 0.38 | Clarification | 0.07 |
| Moderation Advice | 0.38 | - | - |
| Negotiation | 0.38 | - | - |
| Logical Thinking | 0.38 | - | - |
| Gratitude Expression | 0.38 | - | - |
| Helpfulness | 0.38 | - | - |

Table 15: Full distribution of conversational skills in each dataset, *sm*-BST, *sm*-BSTand *sm*-ProsocialDialog, as provided in THANOSBENCH.

- **IP, EX, ER:** This is an automatic metric to measure the empathy of generated responses using a fine-tuned RoBERTa model on Epitome dataset Sharma et al. (2020). The Epitome-based metric assigns one of three values—0 (no empathy), 1 (weak empathy), or 2 (strong empathy)—to the generated response. Epitome is a new conceptual framework for expressing empathy which consists of three communication mechanisms. (1) **Interpretation (IP)**: Expression of acknowledgments or understanding of the interlocutor's emotion or situation; (2) **Exploration (EX)**: Expression of active interest in the interlocutor's situation; (3) **Emotional Reaction** (ER): Expression of emotions such as warmth, compassion, and concern in the interlocutor's situation.

- **diff-IP, diff-EX, diff-ER:** This metric Lee et al. (2022a) measures the difference in IP, EX, ER scores between the human (golden) response and the generated response. This metric quantifies how closely the generated response's empathy level aligns with that of the golden human response. A lower score is better.

| Datasets | List of Conversational Skills |
|---|---|
| *sm*-BST | Clarification, Moderation Advice, Image-Commenting, Preference Elicitation, Confirmation, Humor, Gratitude Expression, Mentoring, Self-disclosure, Echoing, Curiosity, Helpfulness, Reflective Listening, Knowledge Sharing, Image-Sharing, Memory Recall, Immediate Response, Creative Problem Solving, Logical Thinking, Decision-making, Personal Background, Negotiation, Commonsense Understanding, Recommendation, Persona Recall, Knowledge Acquisition, Active Listening, Topic Transition, Cultural Sensitivity, Critical Thinking, Empathy |
| *sm*-PhotoChat | Image-Sharing |
| *sm*-ProsocialDialogue | Clarification, Conflict Resolution, Confirmation, Preference Elicitation, Humor, Avoiding Social Bias, Mentoring, Perspective Taking, Self-disclosure, Honesty, Echoing, Respect for Autonomy, Suggestion, Respecting Dietary Choices, Constructive Feedback, Urgency Recognition, Curiosity, Helpfulness, Reflective Listening, Knowledge Sharing, Immediate Response, Constructive Criticism, Creative Problem Solving, Logical Thinking, Decision-making, Negotiation, Harmlessness, Persuasion, Commonsense Understanding, Recommendation, Encouragement, Persona Recall, Active Listening, Factual Problem Solving, Ethics, Cultural Sensitivity, Conflict Avoidance, Critical Thinking, Empathy |

Table 16: List of conversational skills in each dataset, *sm*-BST, *sm*-PhotoChat, and *sm*-ProsocialDialogue, as provided in THANOSBENCH.

## H.2 FULL RESULTS: SKILL-OF-MIND TASK

| Models | *sm*-BST | | | *sm*-PhotoChat | | | *sm*-ProsocialDialog | | | Average | | |
|---|---|---|---|---|---|---|---|---|---|---|---|---|
| | ExG | SC | All | ExG | SC | All | ExG | SC | All | ExG | SC | All |
| *Proprietary models* | | | | | | | | | | | | |
| Gemini-Pro-1.5 | 42.42 | 23.77 | 8.3 | **50.29** | **34.29** | **22.86** | 66.63 | 17.12 | 7.84 | 62.50 | 18.52 | 8.22 |
| Claude-3.5-Sonnet | 42.57 | 26.42 | **10.94** | 34.86 | 17.14 | 11.43 | 63.34 | 30.5 | 15.61 | 59.49 | 29.59 | 14.79 |
| GPT-4o | 41.43 | **31.32** | 8.68 | 32.57 | 17.14 | 14.29 | **69.8** | **35.97** | **19.06** | **64.58** | **34.85** | **17.34** |
| *Open models* | | | | | | | | | | | | |
| LLaMA-3.1-8B | 25.28 | 13.58 | 1.89 | 45.14 | 28.57 | 17.14 | 52.07 | 35.18 | 6.26 | 47.73 | 31.66 | 5.80 |
| LLaMA-3.1-70B | 37.28 | **25.28** | 6.42 | 38.86 | 17.14 | 11.43 | 60.85 | 35.11 | 11.73 | 56.70 | 33.20 | 10.89 |
| LLaMA-3.1-405B | 32.45 | 24.91 | 5.66 | 33.71 | 14.29 | 8.57 | 64.09 | 27.84 | 12.59 | 58.50 | 27.10 | 11.42 |
| Gemma-2-9B | 35.85 | 21.51 | 5.66 | **45.71** | 37.14 | 22.86 | 60.03 | **40.22** | 13.24 | 55.94 | **37.22** | 12.25 |
| Gemma-2-27B | 33.28 | 18.87 | 6.04 | 43.43 | **42.86** | **25.71** | 64.69 | 35.83 | 13.67 | 59.33 | 33.31 | 12.72 |
| Qwen-2.5-7B | 24.53 | 10.57 | 3.02 | 38.29 | 31.43 | 17.14 | 54.37 | 20.07 | 8.42 | 49.36 | 18.82 | 7.75 |
| Qwen-2.5-72B | **38.11** | 18.87 | **7.17** | 41.14 | 25.71 | 20 | **70.56** | 33.02 | **18.85** | **64.86** | 30.65 | **17.04** |
| THANOS *suite* | | | | | | | | | | | | |
| THANOS-1B | 37.06 | 27.55 | 7.17 | 38.86 | 14.29 | 11.43 | 54.45 | 31.73 | 11.44 | 51.40 | 30.71 | 10.77 |
| THANOS-3B | **41.66** | 22.26 | **7.92** | 40 | 17.14 | 14.29 | 58.76 | 30.07 | 11.44 | 55.69 | 28.58 | 10.95 |
| THANOS-8B | 39.02 | 28.3 | 5.66 | **46.29** | **22.86** | **17.14** | 63.71 | **34.32** | **14.82** | 59.48 | **33.14** | **13.43** |

Table 17: Overall zero-shot performance of the skill-of-mind task in THANOSBENCH. The best-performing model in each group (i.e., proprietary models, open models, and THANOS suite) is highlighted in **bold**.

Table 17 present the full results of skill-of-mind task in THANOSBENCH.

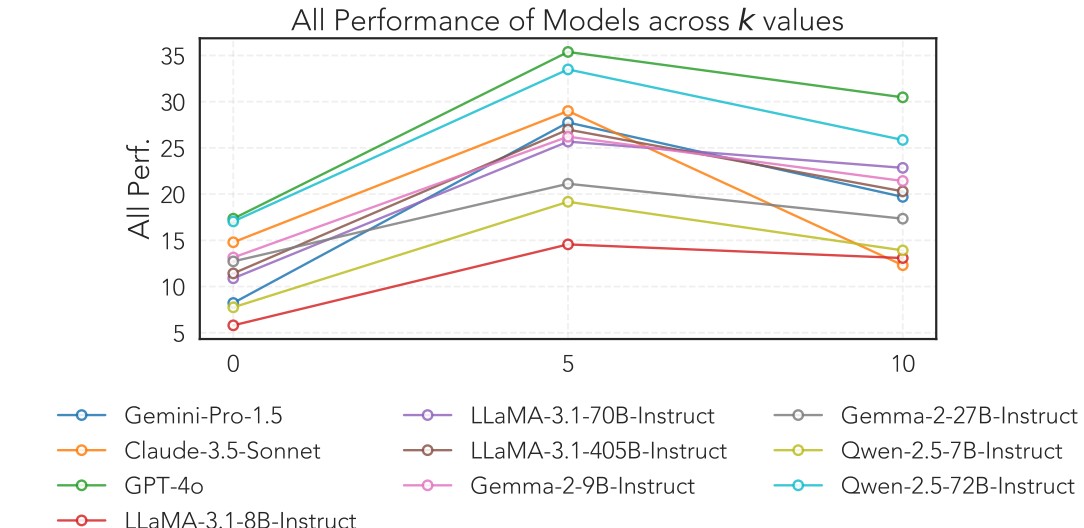

Figure 4: Average zero-shot performance across all metrics in the Skill-of-Mind task in THANOS-BENCH, based on varying the number of hint conversational skills ($k$) provided in the prompt. Full results are shown in Table 18.

### H.3 FULL RESULTS: VARYING NUMBER OF HINT CONVERSATIONAL SKILLS

**Providing hints to LLMs enhances skill-of-mind capabilities, but excessive hints can be detrimental.** We examine whether providing hint conversational skills in the prompt improves an LLM's skill-of-mind capability. As shown in Figure 4, all models exhibit improved performance when provided with $k = 5$ conversational skills (one correct skill along with four distractor skills randomly sampled from a predefined skill set), achieving an average increase (+14.02) across all models. Among them, GPT-4o performs the best, followed closely by Qwen-2.5-72B-Instruct. However, when increasing the number of provided skills from $k = 5$ to $k = 10$, performance declines across all models (-6.21). Notably, Claude-3.5-Sonnet reports a significant performance decrease, falling below even the Qwen-2.5-7B-Instruct. This result suggests that (1) providing an excessive number of hints can confuse LLMs, making it harder for them to correctly identify the appropriate conversational skill. (2) The Skill-of-Mind task inherently involves multiple plausible conversational skills, as it does not provide a "golden" next response. LLMs may infer the existence of multiple valid skills, highlighting the subjective nature of skill-of-mind reasoning and the complexities of social conversation.

### H.4 AUTOMATIC EVALUATION RESULTS ON RESPONSE GENERATION TASK

Table 19 shows the automatic evaluation results on response generation task. Table 20 shows the actual generation result.

### H.5 GRANULARITY OF CONVERSATIONAL SKILL: FINE-GRAINED VS. COARSE-GRAINED

Table 21 presents a performance comparison when training LLaMA-3.1-8B-Instruct on the same dataset, constructed using the same annotation process with GPT-4o-2024-11-20. The dataset consists of 50K randomly sampled dialogues from the Soda dataset, but with different levels of conversational skill granularity. We define coarse-grained skills as those already used in our previous experiments, whereas fine-grained skills are derived by prompting GPT-4o to generate more nuanced distinctions, such as distinguishing between "Empathy" (coarse) and "Empathetic Inquiry" (fine).

Table 21 shows that training THANOS on the coarse-grained dataset leads to significantly better performance compared to the fine-grained dataset. However, this result does not necessarily imply that coarse-grained skills are inherently more effective. The evaluation in THANOSBENCH assesses

| Models | $k$ | sm-BST | | | sm-PhotoChat | | | sm-ProsocialDialog | | | Average | | |
|---|---|---|---|---|---|---|---|---|---|---|---|---|---|
| | | ExG | SC | All | ExG | SC | All | ExG | SC | All | ExG | SC | All |
| *Proprietary models* | | | | | | | | | | | | | |
| Gemini-Pro-1.5 | 0 | 42.42 | 23.77 | 8.3 | 50.29 | 34.29 | 22.86 | 66.63 | 17.12 | 7.84 | 62.50 | 18.52 | 8.22 |
| | 5 | 43.17 | 50.94 | 14.72 | 50.29 | 57.14 | 20 | 70.73 | 61.37 | 30.43 | 65.99 | 59.64 | 27.75 |
| | 10 | 45.36 | 41.13 | 13.21 | 54.29 | 45.71 | 25.71 | 67.22 | 44.96 | 20.79 | 63.53 | 44.38 | 19.70 |
| Claude-3.5-Sonnet | 0 | 42.57 | 26.42 | 10.94 | 34.86 | 17.14 | 11.43 | 63.34 | 30.5 | 15.61 | 59.49 | 29.59 | 14.79 |
| | 5 | 48.98 | 56.6 | 18.49 | 48.57 | 42.86 | 20 | 71.35 | 62.3 | 31.22 | 67.37 | 61.01 | 28.99 |
| | 10 | 44.08 | 44.53 | 15.09 | 39.43 | 25.71 | 14.29 | 38.2 | 21.73 | 11.73 | 39.15 | 25.38 | 12.31 |
| GPT-4o | 0 | 41.43 | 31.32 | 8.68 | 32.57 | 17.14 | 14.29 | 69.8 | 35.97 | 19.06 | 64.58 | 34.85 | 17.34 |
| | 5 | 51.7 | 60.75 | 22.64 | 41.14 | 34.29 | 22.86 | 76.72 | 65.83 | 38.13 | 72.06 | 64.38 | 35.38 |
| | 10 | 50.64 | 52.45 | 21.89 | 43.43 | 34.29 | 22.86 | 74.43 | 54.53 | 32.3 | 70.06 | 53.79 | 30.47 |
| *Open models* | | | | | | | | | | | | | |
| LLaMA-3.1-8B-Instruct | 0 | 25.28 | 13.58 | 1.89 | 45.14 | 28.57 | 17.14 | 52.07 | 35.18 | 6.26 | 47.73 | 31.66 | 5.8 |
| | 5 | 31.4 | 40.38 | 9.06 | 51.43 | 54.29 | 28.57 | 59.87 | 66.19 | 15.25 | 55.23 | 61.89 | 14.56 |
| | 10 | 31.02 | 25.28 | 3.77 | 46.86 | 45.71 | 22.86 | 58.81 | 57.05 | 14.6 | 54.2 | 51.83 | 13.08 |
| LLaMA-3.1-70B-Instruct | 0 | 37.28 | 25.28 | 6.42 | 38.86 | 17.14 | 11.43 | 60.85 | 35.11 | 11.73 | 56.7 | 33.2 | 10.89 |
| | 5 | 41.51 | 52.83 | 14.72 | 38.86 | 28.57 | 20 | 69.8 | 68.42 | 27.91 | 64.72 | 65.15 | 25.68 |
| | 10 | 42.72 | 43.77 | 11.7 | 45.14 | 22.86 | 20 | 68.26 | 56.62 | 25.04 | 63.78 | 53.91 | 22.84 |
| LLaMA-3.1-405B-Instruct | 0 | 32.45 | 24.91 | 5.66 | 33.71 | 14.29 | 8.57 | 64.09 | 27.84 | 12.59 | 58.5 | 27.1 | 11.42 |
| | 5 | 42.79 | 56.98 | 14.34 | 41.14 | 37.14 | 20 | 72.68 | 70.86 | 29.57 | 67.34 | 67.99 | 26.98 |
| | 10 | 42.64 | 45.28 | 12.08 | 38.86 | 20 | 17.14 | 68.82 | 54.32 | 21.94 | 64.09 | 52.19 | 20.3 |
| Gemma-2-9B-Instruct | 0 | 35.85 | 21.51 | 5.66 | 45.71 | 37.14 | 22.86 | 60.27 | 40.22 | 14.32 | 56.14 | 37.22 | 13.14 |
| | 5 | 42.19 | 53.58 | 11.32 | 44.57 | 37.14 | 22.86 | 71.04 | 72.16 | 29.14 | 65.96 | 68.52 | 26.21 |
| | 10 | 40.91 | 45.66 | 10.94 | 48 | 34.29 | 22.86 | 66.53 | 58.42 | 23.38 | 62.13 | 55.92 | 21.42 |
| Gemma-2-27B-Instruct | 0 | 33.28 | 18.87 | 6.04 | 43.43 | 42.86 | 25.71 | 64.69 | 35.83 | 13.67 | 59.33 | 33.31 | 12.72 |
| | 5 | 38.72 | 49.81 | 10.57 | 44 | 37.14 | 17.14 | 67.11 | 62.37 | 23.24 | 62.18 | 59.88 | 21.12 |
| | 10 | 38.26 | 37.36 | 7.55 | 47.43 | 31.43 | 25.71 | 65.68 | 50.07 | 18.99 | 61.01 | 47.69 | 17.34 |
| Qwen-2.5-7B-Instruct | 0 | 24.53 | 10.57 | 3.02 | 38.29 | 31.43 | 17.14 | 54.37 | 20.07 | 8.42 | 49.36 | 18.82 | 7.75 |
| | 5 | 27.77 | 37.36 | 5.28 | 37.71 | 45.71 | 20 | 63.45 | 58.92 | 21.8 | 57.33 | 55.27 | 19.17 |
| | 10 | 27.4 | 29.81 | 4.91 | 42.86 | 42.86 | 17.14 | 59.61 | 41.94 | 15.54 | 54.21 | 40.06 | 13.91 |
| Qwen-2.5-72B-Instruct | 0 | 38.11 | 18.87 | 7.17 | 41.14 | 25.71 | 20 | 70.56 | 33.02 | 18.85 | 64.86 | 30.65 | 17.04 |
| | 5 | 42.72 | 55.09 | 13.58 | 50.86 | 34.29 | 20 | 77.6 | 67.63 | 37.63 | 71.57 | 64.97 | 33.49 |
| | 10 | 41.96 | 41.51 | 13.96 | 36.57 | 25.71 | 8.57 | 74.36 | 53.38 | 28.56 | 68.5 | 50.95 | 25.86 |

Table 18: Overall zero-shot performance on the Skill-of-Mind task in THANOSBENCH, varying the number of provided conversational skills ($k$) used as hints in the prompt for the models.

skill-of-mind capability based on coarse-grained conversational skills, which may introduce an inherent bias favoring the coarse-level model. To enable a more accurate assessment of fine-grained skills, constructing a fine-grained benchmark is necessary, which we leave for future work. Interestingly, we also observe that THANOS-8B-Coarse outperforms THANOS-8B despite being trained on only 50K dialogues. This finding suggests that high-quality datasets play a more critical role than dataset size, aligning with the growing emphasis on data-centric AI Lee et al. (2024c); Kim et al. (2024c); Muennighoff et al. (2025).

## H.6 INVESTIGATING METAL-CONVERSATIONAL SKILL

We forcibly inject conversational skills (without explanation) into the prompt template during response generation. As shown in Figure 5, the "Rhetoric" skill enhances performance across all five evaluation criteria, suggesting that it functions as a meta-conversational skill. However, the "Immediate Response" skill significantly degrades performance, indicating that responding as quickly as possible without reasoning is not always beneficial; rather, understanding the conversational context is crucial. We plan to expand these experiments to cluster skills in future work.

| Models | BST | | | DailyDialog | | | Mutual | | | Empathy | | | ProsocialDialog | | | Average | | |
|---|---|---|---|---|---|---|---|---|---|---|---|---|---|---|---|---|---|---|
| | B-1 | B-2 | R-L | B-1 | B-2 | R-L | B-1 | B-2 | R-L | B-1 | B-2 | R-L | B-1 | B-2 | R-L | B-1 | B-2 | R-L |
| *Proprietary models* | | | | | | | | | | | | | | | | | | |
| Gemini-Pro-1.5 | 0.193 | 0.0872 | 0.1462 | 0.2407 | 0.1296 | 0.2277 | 0.1578 | 0.071 | 0.1652 | 0.2062 | 0.0824 | 0.1617 | 0.223 | 0.0873 | 0.146 | 0.2041 | 0.0915 | 0.1694 |
| + THANOS-1B | 0.1758 | 0.0676 | 0.1346 | 0.2053 | 0.0979 | 0.1753 | 0.1863 | 0.0781 | 0.1445 | 0.1741 | 0.0637 | 0.1415 | 0.2681 | 0.1006 | 0.1581 | 0.2019 | 0.0816 | 0.1508 |
| + THANOS-3B | 0.1745 | 0.0699 | 0.1329 | 0.2071 | 0.0959 | 0.168 | 0.21 | 0.0882 | 0.1518 | 0.174 | 0.0644 | 0.1411 | 0.2668 | 0.0995 | 0.1586 | 0.2065 | 0.0836 | 0.1505 |
| + THANOS-8B | 0.1736 | 0.0661 | 0.1312 | 0.2106 | 0.0968 | 0.1863 | 0.1998 | 0.081 | 0.1494 | 0.1762 | 0.0678 | 0.1409 | 0.2722 | 0.1036 | 0.1597 | 0.2065 | 0.0831 | 0.1535 |
| Gemini-2.0-Flash | 0.1879 | 0.0722 | 0.1366 | 0.2235 | 0.1181 | 0.2183 | 0.1297 | 0.06 | 0.1583 | 0.1689 | 0.0663 | 0.1494 | 0.1521 | 0.0591 | 0.1413 | 0.1724 | 0.0751 | 0.1608 |
| + THANOS-1B | 0.1792 | 0.0669 | 0.1314 | 0.2211 | 0.106 | 0.1896 | 0.1615 | 0.0662 | 0.1403 | 0.1838 | 0.0691 | 0.1361 | 0.1822 | 0.0671 | 0.1506 | 0.1856 | 0.0751 | 0.1496 |
| + THANOS-3B | 0.1768 | 0.066 | 0.1306 | 0.2199 | 0.0998 | 0.1704 | 0.16 | 0.0677 | 0.1467 | 0.1928 | 0.0732 | 0.145 | 0.1793 | 0.0659 | 0.1492 | 0.1858 | 0.0745 | 0.1484 |
| + THANOS-8B | 0.1768 | 0.0651 | 0.1306 | 0.2293 | 0.1119 | 0.1872 | 0.1693 | 0.072 | 0.1451 | 0.1807 | 0.0662 | 0.1357 | 0.1847 | 0.069 | 0.154 | 0.1882 | 0.0768 | 0.1505 |
| Claude-3.5-Sonnet | 0.1714 | 0.0691 | 0.1427 | 0.2132 | 0.1049 | 0.191 | 0.2503 | 0.1106 | 0.1735 | 0.1669 | 0.0617 | 0.1428 | 0.2734 | 0.1025 | 0.1625 | 0.2150 | 0.0898 | 0.1625 |
| + THANOS-1B | 0.157 | 0.0601 | 0.134 | 0.1991 | 0.0947 | 0.181 | 0.2314 | 0.0978 | 0.1557 | 0.1548 | 0.0533 | 0.1311 | 0.2648 | 0.0969 | 0.1603 | 0.2014 | 0.0806 | 0.1524 |
| + THANOS-3B | 0.1553 | 0.0601 | 0.1326 | 0.1956 | 0.0951 | 0.1782 | 0.2328 | 0.0946 | 0.1555 | 0.1552 | 0.0542 | 0.134 | 0.2638 | 0.098 | 0.1622 | 0.2005 | 0.0804 | 0.1525 |
| + THANOS-8B | 0.1592 | 0.0606 | 0.139 | 0.2 | 0.099 | 0.1825 | 0.2358 | 0.0984 | 0.1534 | 0.1546 | 0.0544 | 0.1315 | 0.2649 | 0.0973 | 0.1628 | 0.2029 | 0.0819 | 0.1538 |
| GPT-4o | 0.1619 | 0.0586 | 0.1355 | 0.2071 | 0.1004 | 0.1922 | 0.2291 | 0.0941 | 0.159 | 0.1747 | 0.0655 | 0.1485 | 0.2657 | 0.0963 | 0.1624 | 0.2077 | 0.0830 | 0.1595 |
| + THANOS-1B | 0.1388 | 0.0512 | 0.1235 | 0.162 | 0.0716 | 0.147 | 0.2243 | 0.0876 | 0.1508 | 0.155 | 0.0557 | 0.1354 | 0.2491 | 0.0861 | 0.1575 | 0.1858 | 0.0704 | 0.1454 |
| + THANOS-3B | 0.1398 | 0.0506 | 0.1282 | 0.1715 | 0.077 | 0.1695 | 0.2191 | 0.0846 | 0.1439 | 0.152 | 0.0512 | 0.1355 | 0.251 | 0.0874 | 0.1583 | 0.1867 | 0.0702 | 0.1471 |
| + THANOS-8B | 0.1462 | 0.053 | 0.1265 | 0.176 | 0.076 | 0.1603 | 0.2325 | 0.0973 | 0.1496 | 0.1561 | 0.0554 | 0.1345 | 0.252 | 0.0864 | 0.1578 | 0.1926 | 0.0736 | 0.1457 |
| *Open models* | | | | | | | | | | | | | | | | | | |
| LLaMA-3.1-8B-Instruct | 0.147 | 0.0468 | 0.1184 | 0.17 | 0.0625 | 0.1414 | 0.2245 | 0.0834 | 0.1394 | 0.157 | 0.0525 | 0.1341 | 0.2326 | 0.0792 | 0.1467 | 0.1862 | 0.0649 | 0.1360 |
| + THANOS-1B | 0.1195 | 0.0365 | 0.1099 | 0.144 | 0.0563 | 0.1311 | 0.1906 | 0.0651 | 0.1251 | 0.1374 | 0.0429 | 0.1191 | 0.2247 | 0.0722 | 0.145 | 0.1632 | 0.0546 | 0.1260 |
| + THANOS-3B | 0.1198 | 0.0324 | 0.1115 | 0.142 | 0.0475 | 0.1389 | 0.2029 | 0.0741 | 0.1326 | 0.138 | 0.0396 | 0.1237 | 0.2228 | 0.0724 | 0.1432 | 0.1651 | 0.0532 | 0.1300 |
| + THANOS-8B | 0.1204 | 0.0341 | 0.1054 | 0.1482 | 0.0551 | 0.131 | 0.2014 | 0.0708 | 0.1374 | 0.1403 | 0.046 | 0.1239 | 0.2236 | 0.0728 | 0.1445 | 0.1668 | 0.0558 | 0.1284 |
| LLaMA-3.1-70B-Instruct | 0.1455 | 0.053 | 0.1274 | 0.1771 | 0.0775 | 0.1576 | 0.2309 | 0.0975 | 0.1553 | 0.1541 | 0.0523 | 0.1364 | 0.2501 | 0.092 | 0.1616 | 0.1915 | 0.0745 | 0.1477 |
| + THANOS-1B | 0.1222 | 0.0367 | 0.1103 | 0.1495 | 0.0594 | 0.1463 | 0.1937 | 0.0732 | 0.1374 | 0.1353 | 0.0453 | 0.12 | 0.2429 | 0.087 | 0.1569 | 0.1687 | 0.0603 | 0.1342 |
| + THANOS-3B | 0.1257 | 0.0423 | 0.1162 | 0.144 | 0.0578 | 0.1468 | 0.2061 | 0.0807 | 0.1397 | 0.1345 | 0.0412 | 0.121 | 0.2393 | 0.0856 | 0.1554 | 0.1699 | 0.0615 | 0.1358 |
| + THANOS-8B | 0.1277 | 0.0425 | 0.1198 | 0.1593 | 0.0674 | 0.1549 | 0.2085 | 0.0834 | 0.1467 | 0.1385 | 0.0446 | 0.1227 | 0.2436 | 0.0875 | 0.1576 | 0.1755 | 0.0651 | 0.1403 |
| LLaMA-3.1-405B-Instruct | 0.1571 | 0.0545 | 0.1301 | 0.1994 | 0.0957 | 0.183 | 0.2391 | 0.1032 | 0.1571 | 0.1601 | 0.0552 | 0.1365 | 0.2566 | 0.0963 | 0.1626 | 0.2025 | 0.0810 | 0.1539 |
| + THANOS-1B | 0.1232 | 0.0397 | 0.1119 | 0.1507 | 0.0647 | 0.1469 | 0.1909 | 0.0691 | 0.13 | 0.1354 | 0.0436 | 0.122 | 0.2355 | 0.0845 | 0.1569 | 0.1671 | 0.0603 | 0.1335 |
| + THANOS-3B | 0.126 | 0.0414 | 0.1139 | 0.1485 | 0.0652 | 0.1473 | 0.1953 | 0.0738 | 0.1411 | 0.1372 | 0.0425 | 0.124 | 0.235 | 0.0864 | 0.1582 | 0.1684 | 0.0619 | 0.1369 |
| + THANOS-8B | 0.1275 | 0.0412 | 0.1203 | 0.155 | 0.0646 | 0.1526 | 0.1983 | 0.0775 | 0.1379 | 0.1391 | 0.044 | 0.1258 | 0.2354 | 0.0849 | 0.1559 | 0.1711 | 0.0624 | 0.1385 |
| Gemma-2-9B-Instruct | 0.1251 | 0.047 | 0.1177 | 0.1783 | 0.0881 | 0.1746 | 0.0736 | 0.0303 | 0.115 | 0.1276 | 0.048 | 0.1368 | 0.1816 | 0.0629 | 0.1485 | 0.1372 | 0.0553 | 0.1385 |
| + THANOS-1B | 0.1389 | 0.0474 | 0.1087 | 0.1666 | 0.0743 | 0.1383 | 0.0787 | 0.0329 | 0.0896 | 0.1484 | 0.0546 | 0.1221 | 0.165 | 0.0565 | 0.1359 | 0.1395 | 0.0531 | 0.1189 |
| + THANOS-3B | 0.1586 | 0.0553 | 0.1194 | 0.1908 | 0.0841 | 0.1527 | 0.0813 | 0.0333 | 0.092 | 0.1501 | 0.051 | 0.1229 | 0.17 | 0.0597 | 0.1381 | 0.1502 | 0.0567 | 0.1250 |
| + THANOS-8B | 0.1457 | 0.0516 | 0.1137 | 0.1877 | 0.0852 | 0.1559 | 0.0833 | 0.0332 | 0.0925 | 0.1499 | 0.0542 | 0.1215 | 0.1756 | 0.0618 | 0.1416 | 0.1484 | 0.0572 | 0.1250 |
| Gemma-2-27B-Instruct | 0.0889 | 0.0362 | 0.1189 | 0.1353 | 0.0703 | 0.1872 | 0.0465 | 0.0207 | 0.1143 | 0.0632 | 0.0241 | 0.1118 | 0.1024 | 0.0361 | 0.1258 | 0.0873 | 0.0375 | 0.1316 |
| + THANOS-1B | 0.1356 | 0.0449 | 0.111 | 0.1656 | 0.0741 | 0.1482 | 0.1108 | 0.0437 | 0.1162 | 0.1297 | 0.0471 | 0.1197 | 0.1376 | 0.047 | 0.1341 | 0.1359 | 0.0514 | 0.1258 |
| + THANOS-3B | 0.1458 | 0.0502 | 0.1123 | 0.1765 | 0.0783 | 0.1563 | 0.1129 | 0.0433 | 0.1139 | 0.1307 | 0.0452 | 0.1222 | 0.142 | 0.0501 | 0.1371 | 0.1416 | 0.0534 | 0.1284 |
| + THANOS-8B | 0.1355 | 0.0478 | 0.1115 | 0.1771 | 0.0766 | 0.1478 | 0.1039 | 0.0412 | 0.1225 | 0.1231 | 0.0455 | 0.115 | 0.1406 | 0.0493 | 0.1351 | 0.1360 | 0.0521 | 0.1264 |
| Qwen-2.5-7B-Instruct | 0.1681 | 0.0634 | 0.1309 | 0.2003 | 0.0906 | 0.1702 | 0.1883 | 0.0722 | 0.1462 | 0.1753 | 0.0584 | 0.1329 | 0.2134 | 0.0687 | 0.1426 | 0.1891 | 0.0707 | 0.1446 |
| + THANOS-1B | 0.1374 | 0.0464 | 0.1175 | 0.1524 | 0.0615 | 0.134 | 0.2059 | 0.0762 | 0.1361 | 0.1523 | 0.0478 | 0.1234 | 0.2458 | 0.0788 | 0.1436 | 0.1788 | 0.0621 | 0.1309 |
| + THANOS-3B | 0.1369 | 0.0467 | 0.1159 | 0.1586 | 0.0655 | 0.1521 | 0.2075 | 0.0787 | 0.1341 | 0.156 | 0.0497 | 0.1295 | 0.2479 | 0.08 | 0.1436 | 0.1814 | 0.0641 | 0.1350 |
| + THANOS-8B | 0.1383 | 0.0466 | 0.1184 | 0.1606 | 0.0654 | 0.1542 | 0.2178 | 0.0873 | 0.1363 | 0.1562 | 0.0472 | 0.125 | 0.2506 | 0.0803 | 0.1458 | 0.1847 | 0.0654 | 0.1359 |
| Qwen-2.5-72B-Instruct | 0.1723 | 0.0662 | 0.1428 | 0.2026 | 0.0967 | 0.1883 | 0.2495 | 0.1 | 0.165 | 0.1694 | 0.0594 | 0.1493 | 0.249 | 0.0917 | 0.1643 | 0.2086 | 0.0828 | 0.1619 |
| + THANOS-1B | 0.1504 | 0.0559 | 0.1323 | 0.1745 | 0.0822 | 0.1626 | 0.2226 | 0.0861 | 0.1516 | 0.1581 | 0.053 | 0.1414 | 0.2569 | 0.0944 | 0.1683 | 0.1925 | 0.0743 | 0.1512 |
| + THANOS-3B | 0.1495 | 0.0562 | 0.1303 | 0.1736 | 0.0802 | 0.1691 | 0.235 | 0.0934 | 0.1592 | 0.1603 | 0.0558 | 0.1444 | 0.2553 | 0.0941 | 0.1664 | 0.1947 | 0.0759 | 0.1539 |
| + THANOS-8B | 0.1532 | 0.0555 | 0.1306 | 0.1777 | 0.084 | 0.1693 | 0.2328 | 0.0947 | 0.1582 | 0.1609 | 0.0562 | 0.1427 | 0.2562 | 0.0945 | 0.1678 | 0.1962 | 0.0770 | 0.1537 |

Table 19: Automatic evaluation results of response generation task in THANOSBENCH. B-1/2/4 refer to BLEU-1/2/4 Papineni et al. (2002), and R-L refers to ROUGE-L Lin (2004) for simplicity.

# I  META-EVALUATION RESULTS ON MULTI-TURN CONVERSATION DATASETS

In this work, instead of conducting a human evaluation, we leverage LLM-as-a-Judge Zheng et al. (2023); Liu et al. (2023b) to assess response quality in social conversations. To ensure the reliability and robustness of this approach, we evaluate LLMs as proxy human evaluators on seven meta-evaluation multi-turn dialogue datasets: DailyDialog-Zhao & ConvAI2-Zhao Zhao et al. (2020), ConvAI2-USR & TopicalChat-USR Mehri & Eskenazi (2020), DailyDialog-GRADE & ConvAI2-GRADE Huang et al. (2020), and DailyDialog-Ours. We assess the alignment between LLM-predicted scores and human annotations provided in these datasets. Six of these datasets (except DailyDialog-Ours) contain annotations on a 1-5 rating scale, making them suitable for direct assessment. For pairwise comparison, we use DailyDialog-Ours dataset encompassing human preferences obtained from our head-to-head evaluation experiment (Table 6).

For evaluator LMs, we test multiple models, including LLaMA-2-Chat-7B Touvron et al. (2023), LLaMA-3.1-8B-Instruct, LLaMA-3.2-1,3B-Instruct AI@Meta (2024), Prometheus-2-7B Kim et al. (2024b), GPT-4o-mini-2024-07-18, and GPT-4o-2024-11-20 Hurst et al. (2024).

## I.1  META-EVALUATION RESULTS OF DIRECT ASSESSMENT

We measure Pearson, Spearman, and Kendall-Tau correlations between evaluator LMs and human reference evaluators across six meta-evaluation datasets. As shown in Table 23, Table 24, and Table 25, general-purpose LLMs (e.g., LLaMA-series) do not perform well as evaluator LMs in social

👨‍👩 **Social Context Information**:
Two speakers are communicate with each other.

💬 **Dialogue**:
**Speaker A:** I'll be willing to come and talk about the financing of our imports.
**Speaker B:** It can be solved by drawing a draft on us at 90 days sight.
**Speaker A:** What about a draft at 120 days sight?
**Speaker B:** All right. But we demand the draft be accepted by a bank acceptable to us.
**Speaker A:** A bank's acceptance will add to the costs of our imports. You can rest assured that we will accept the draft and honour it when it comes due.
**Speaker B:** Then we will be in a position to arrange for a loan from our bank. You know we exports would like to have our investments returned as early as possible.
**Speaker A:** I hope this transaction will pave the way for further business relations between our two countries.

💡 **Inferred Skill-of-Mind (THANOS-3B)**:
• **Explanation**: By using conflict avoidance, I can express satisfaction in the agreement's feasibility while keeping the negotiation dynamic and hopeful. This shows a readiness to maintain a cooperative relationship, despite acknowledging the challenges in securing a draft at the desired time.
• **Conversational Skill**: Conflict Avoidance.

**Base Next Response**:
**Speaker B:** We share the same hope, and we are willing to work together to achieve that goal.

**THANOS-3B Next Response**:
**Speaker B:** I'm glad we could come to a mutually beneficial agreement, and I share your enthusiasm for the potential of this transaction to foster stronger business ties between our countries, which could lead to more collaborative and profitable ventures in the future.

Table 20: An actual example generated by LLaMA-3.1-70B-Instruct in THANOSBENCH.

| Models | ExG | SC | All |
|---|---|---|---|
| THANOS-8B-Fine | 60.99 | 6.80 | 2.54 |
| THANOS-8B-Coarse | **64.06** | **32.60** | **16.09** |

Table 21: Comparison of zero-shot performance across different granularities of conversational skills (fine-grained vs. coarse-grained) in the skill-of-mind task within THANOSBENCH.

| | sm-BST | | | | | sm-PhotoChat | | | | | sm-ProsocialDialog | | | | | Average | | | | |
|---|---|---|---|---|---|---|---|---|---|---|---|---|---|---|---|---|---|---|---|---|
| Models | SS | PT | MT | NM | IC | SS | PT | MT | NM | IC | SS | PT | MT | NM | IC | SS | PT | MT | NM | IC |
| *Proprietary models* | | | | | | | | | | | | | | | | | | | | |
| Gemini-Pro-1.5 | 22.26 | 24.91 | 51.70 | 65.66 | 47.55 | 28.57 | 37.14 | 57.14 | 80.00 | 48.57 | 40.72 | 45.83 | 92.52 | 76.98 | 77.12 | 37.57 | 42.37 | 85.38 | 75.27 | 71.89 |
| Claude-3.5-Sonnet | 18.87 | 28.68 | 55.09 | 71.32 | 38.87 | 20.00 | 17.14 | 37.14 | 62.86 | 37.14 | 29.64 | 42.66 | 90.14 | 81.51 | 72.73 | 27.75 | 39.94 | 83.55 | 79.53 | 66.69 |
| GPT-4o | 19.62 | 24.91 | 48.68 | 73.21 | 40.75 | 20.00 | 20.00 | 34.29 | 60.00 | 28.57 | 45.25 | 48.13 | 90.94 | 84.68 | 80.00 | 40.71 | 43.91 | 83.14 | 82.37 | 72.78 |
| *Open models* | | | | | | | | | | | | | | | | | | | | |
| LLaMA-3.1-8B-Instruct | 4.15 | 12.08 | 29.81 | 60.75 | 19.62 | 25.71 | 34.29 | 45.71 | 77.14 | 42.86 | 11.58 | 29.93 | 81.51 | 78.27 | 59.06 | 10.71 | 27.22 | 72.66 | 75.50 | 52.54 |
| LLaMA-3.1-70B-Instruct | 11.70 | 23.02 | 49.06 | 66.42 | 36.23 | 17.14 | 22.86 | 45.71 | 71.43 | 37.14 | 24.46 | 39.42 | 88.27 | 81.37 | 70.72 | 22.31 | 36.51 | 81.24 | 78.82 | 64.62 |
| LLaMA-3.1-405B-Instruct | 10.19 | 17.36 | 41.51 | 64.53 | 28.68 | 17.14 | 17.14 | 34.29 | 65.71 | 34.29 | 27.12 | 45.83 | 90.65 | 83.81 | 73.02 | 24.26 | 40.77 | 81.78 | 80.41 | 65.27 |
| Gemma-2-9B-Instruct | 13.21 | 23.77 | 41.89 | 64.53 | 35.85 | 28.57 | 31.43 | 48.57 | 74.29 | 45.71 | 27.99 | 36.69 | 91.29 | 78.99 | 66.40 | 25.68 | 34.56 | 82.66 | 76.63 | 61.18 |
| Gemma-2-27B-Instruct | 12.08 | 17.74 | 44.53 | 60.00 | 32.08 | 34.29 | 34.29 | 48.57 | 60.00 | 40.00 | 36.91 | 44.60 | 90.94 | 76.83 | 74.17 | 32.96 | 40.18 | 82.78 | 73.85 | 66.86 |
| Qwen-2.5-7B-Instruct | 6.04 | 9.43 | 25.28 | 56.98 | 24.91 | 20.00 | 22.86 | 37.14 | 77.14 | 34.29 | 24.75 | 28.92 | 78.49 | 76.98 | 62.73 | 21.72 | 25.74 | 69.29 | 73.85 | 56.21 |
| Qwen-2.5-72B-Instruct | 16.23 | 24.15 | 47.92 | 65.66 | 36.60 | 22.86 | 28.57 | 42.86 | 74.29 | 37.14 | 45.32 | 51.15 | 92.73 | 83.88 | 79.71 | 40.30 | 46.45 | 84.67 | 80.83 | 72.07 |
| THANOS *suite* | | | | | | | | | | | | | | | | | | | | |
| THANOS-1B | 14.34 | 23.40 | 43.02 | 69.06 | 35.47 | 14.29 | 20.00 | 40.00 | 77.14 | 42.86 | 23.67 | 32.30 | 81.80 | 75.61 | 58.85 | 22.01 | 30.65 | 74.85 | 74.62 | 54.85 |
| THANOS-3B | 21.13 | 27.55 | 51.70 | 68.30 | 39.62 | 14.29 | 20.00 | 48.57 | 77.14 | 40.00 | 26.83 | 37.70 | 85.54 | 79.35 | 64.39 | 25.68 | 35.74 | 79.47 | 77.57 | 60.00 |
| THANOS-8B | 14.34 | 29.06 | 47.92 | 63.77 | 40.00 | 25.71 | 34.29 | 51.43 | 77.14 | 42.86 | 30.86 | 43.81 | 90.65 | 81.58 | 71.65 | 28.17 | 41.30 | 83.14 | 78.70 | 66.09 |

Table 22: Full results of the explanation generation task in THANOSBENCH.

dialogue evaluation. In contrast, three specialized models (i.e., Prometheus-2-7B, GPT-4o-mini, GPT-4o) exhibit significantly better performance. Among these, GPT-4o achieves the highest correlation across all datasets, followed by GPT-4o-mini as the second-best model. Based on these results, we initially selected GPT-4o as our evaluator LM. However, considering computational efficiency and cost constraints, we opted for GPT-4o-mini for our large-scale experiments. In future work, if we secure a sufficient budget, we plan to run GPT-4o for further evaluation.

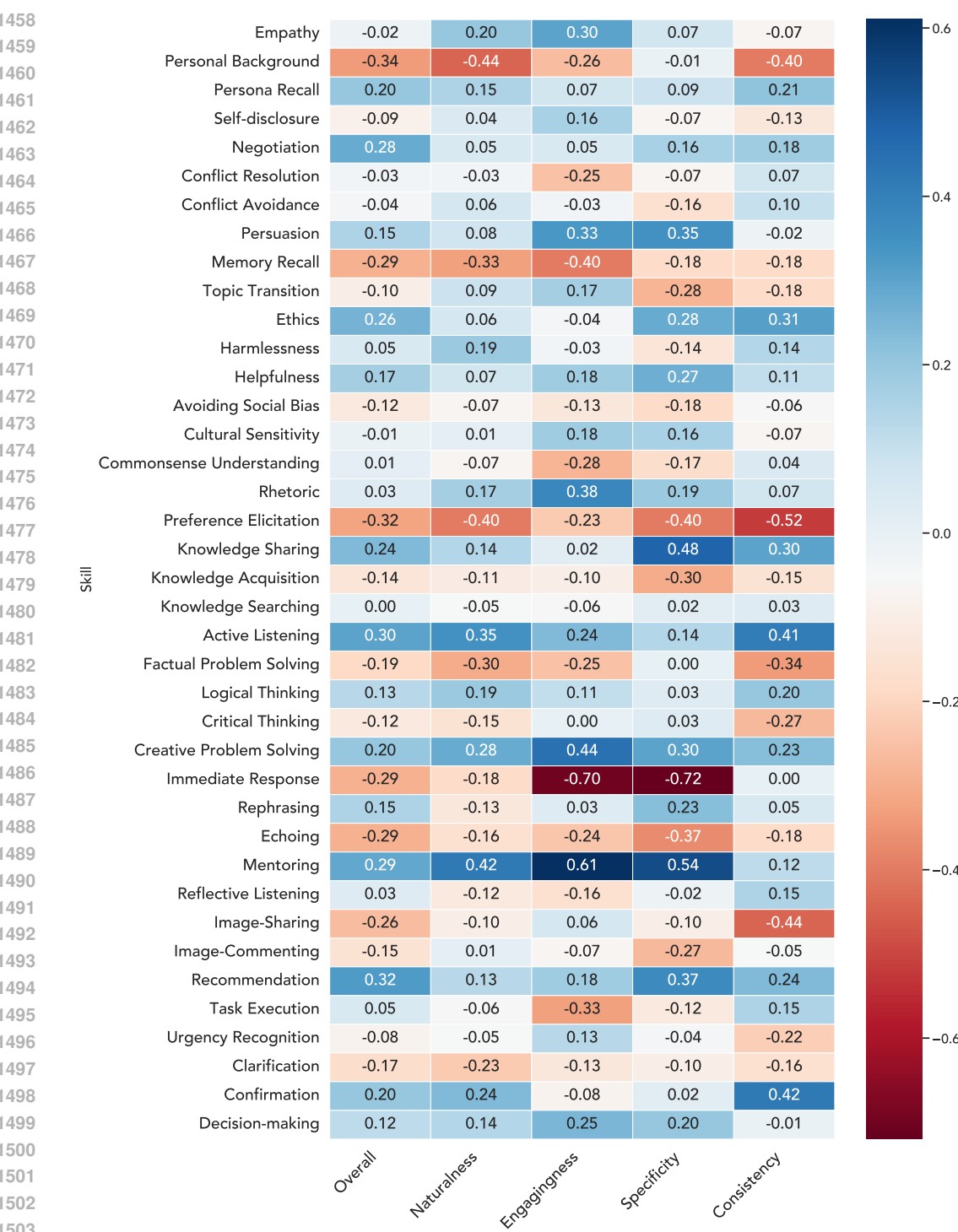

Figure 5: Performance improvements (absolute values) of LLaMA-3.1-70B-Instruct on the response generation task in THANOSBENCH when a conversational skill is forcibly injected, regardless of the dialogue context (direct assessment evaluation setting).

| Evaluator LM | DailyDialog GRADE | DailyDialog Zhao | ConvAI2 GRADE | ConvAI2 USR | ConvAI2 Zhao | TopicalChat USR | Avg |
|---|---|---|---|---|---|---|---|
| LLaMA-2-Chat 7B | 0.103 | 0.054 | 0.089 | 0.029 | 0.061 | 0.017 | 0.059 |
| LLaMA-3.2-1B | 0.019 | 0.040 | -0.015 | 0.130 | -0.035 | -0.018 | 0.020 |
| LLaMA-3.2-3B | 0.036 | 0.095 | 0.053 | -0.021 | 0.081 | 0.095 | 0.056 |
| LLaMA-3.1-8B | 0.165 | 0.326 | 0.245 | 0.143 | 0.378 | 0.218 | 0.246 |
| Prometheus-2-7B | 0.433 | 0.530 | 0.418 | 0.476 | 0.577 | 0.503 | 0.489 |
| GPT-4o-mini | 0.449 | 0.589 | 0.597 | 0.526 | 0.599 | 0.602 | 0.560 |
| GPT-4o | **0.508** | **0.638** | **0.607** | **0.565** | **0.643** | **0.665** | **0.604** |

Table 23: **Pearson correlations** between evaluator LMs and human reference evaluators in direct assessment in six meta-evaluation datasets. Results for Prometheus-2-7B Kim et al. (2024b), GPT-4o-mini, and GPT-4o are statistically significant with $p < 1e\text{-}5$. The best and second-best performances are marked in **bold** and underline, respectively.

| Evaluator LM | DailyDialog GRADE | DailyDialog Zhao | ConvAI2 GRADE | ConvAI2 USR | ConvAI2 Zhao | TopicalChat USR | Avg |
|---|---|---|---|---|---|---|---|
| LLaMA-2-Chat 7B | 0.104 | 0.039 | 0.060 | -0.005 | 0.052 | 0.022 | 0.045 |
| LLaMA-3.2-1B | 0.010 | -0.004 | -0.051 | 0.105 | -0.035 | -0.001 | 0.004 |
| LLaMA-3.2-3B | 0.020 | 0.061 | 0.015 | -0.061 | 0.042 | 0.059 | 0.023 |
| LLaMA-3.1-8B | 0.132 | 0.271 | 0.198 | 0.125 | 0.322 | 0.160 | 0.201 |
| Prometheus-2-7B | 0.346 | 0.445 | 0.334 | 0.380 | 0.481 | 0.382 | 0.395 |
| GPT-4o-mini | 0.356 | 0.534 | **0.469** | 0.442 | 0.552 | 0.503 | 0.476 |
| GPT-4o | **0.378** | **0.545** | 0.456 | **0.450** | **0.558** | **0.547** | **0.489** |

Table 24: **Spearman correlations** between evaluator LMs and human reference evaluators in direct assessment in six meta-evaluation datasets. Results for Prometheus-2-7B Kim et al. (2024b), GPT-4o-mini, and GPT-4o are statistically significant with $p < 1e - 5$. The best and second-best performances are marked in **bold** and underline, respectively.

| Evaluator LM | DailyDialog GRADE | DailyDialog Zhao | ConvAI2 GRADE | ConvAI2 USR | ConvAI2 Zhao | TopicalChat USR | Avg |
|---|---|---|---|---|---|---|---|
| LLaMA-2-Chat 7B | 0.128 | 0.047 | 0.074 | -0.007 | 0.063 | 0.027 | 0.055 |
| LLaMA-3.2-1B | 0.013 | -0.003 | -0.062 | 0.128 | -0.043 | 0.000 | 0.005 |
| LLaMA-3.2-3B | 0.025 | 0.073 | 0.018 | -0.073 | 0.050 | 0.070 | 0.027 |
| LLaMA-3.1-8B | 0.167 | 0.329 | 0.245 | 0.151 | 0.385 | 0.193 | 0.245 |
| Prometheus-2-7B | 0.460 | 0.570 | 0.433 | 0.498 | 0.619 | 0.496 | 0.513 |
| GPT-4o-mini | 0.474 | 0.672 | **0.606** | 0.569 | 0.700 | 0.656 | 0.613 |
| GPT-4o | **0.502** | **0.690** | 0.602 | **0.576** | **0.709** | **0.708** | **0.631** |

Table 25: **Kendall-Tau correlations** between evaluator LMs and human reference evaluators in direct assessment in six meta-evaluation datasets. Results for Prometheus-2-7B Kim et al. (2024b), GPT-4o-mini, and GPT-4o are statistically significant with $p < 1e - 5$. The best and second-best performances are marked in **bold** and underline, respectively.

## I.2 META-EVALUATION RESULTS OF PAIRWISE COMPARISON

We assess the correlation between evaluator LMs and human evaluators by measuring accuracy—specifically, the proportion of cases where the predicted preference aligns with human preference. For this experiment, we evaluate only the three highest-performing LLMs (in Appendix I.1): Prometheus-2-7B, GPT-4o-mini, and GPT-4o. As shown in Table 26, GPT-4o achieves the highest accuracy, followed by GPT-4o-mini as the second-best model. Given budget constraints, we adopt GPT-4o-mini as the evaluator LM for our experiments.

## J HUMAN EVALUATION QUESTIONNAIRE

This section presents the list of questions and multiple-choice options used for the human ratings represented in Section 2.

### J.1 HUMAN RATINGS

- **Relevance:** How relevant is the given explanation to the current dialogue situation and the social context?

  **Options:** 1: Not at all / 2: A little / 3: Somewhat / 4: A lot

- **Plausibility:** Does the given explanation seem plausible, as if a human would think in a real-world scenario?

| Evaluator LM | DailyDialog-Ours |
|---|---|
| Prometheus-2-7B | 58.71 |
| GPT-4o-mini | 63.50 |
| GPT-4o | **65.83** |

Table 26: Accuracy between evaluator LMs and human reference evaluators in pairwise comparison in DailyDialog-Ours dataset. The best and second-best performances are marked in **bold** and underline.

**Options:** 1: Not at all / 2: A little / 3: Somewhat / 4: A lot

- **Understanding:** How does the given explanation demonstrate understanding of the current dialogue situation and the social context?

  **Options:** 1: Not at all / 2: A little / 3: Somewhat / 4: A lot

- **Skill Alignment:** Does the selected conversational skill align well with the provided explanation?

  **Options:** 1: Not at all / 2: A little / 3: Somewhat / 4: A lot

- **Skill Adequacy:** Do the conversational skills currently used seem appropriate for generating a suitable response in the upcoming turn?

  **Options:** 1: Not at all / 2: A little / 3: Somewhat / 4: A lot

### J.2 HEAD-TO-HEAD COMPARISON

- **Naturalness:** Which response is more natural?

  **Options:** Definitely A / Slightly A / Slightly B / Definitely B

- **Consistent:** Which response is more consistent?

  **Options:** Definitely A / Slightly A / Slightly B / Definitely B

- **Specificity:** Which response is more specific?

  **Options:** Definitely A / Slightly A / Slightly B / Definitely B

- **Engagingness:** Which response is more engaging?

  **Options:** Definitely A / Slightly A / Slightly B / Definitely B

- **Overall:** Which response do you like more overall?

  **Options:** Definitely A / Slightly A / Slightly B / Definitely B

### K HUMAN EVALUATION SYSTEM

We show a screenshot of the human evaluation system in Figure 6 and Figure 7. We implement this system using Label Studio Tkachenko et al. (2020-2022).

### L DETAILS OF HUMAN EVALUATION

We recruited 15 individuals, unknown to us, who are either graduate or undergraduate students. Prior to participating in the experiment, they were provided with comprehensive instruction on the task, an overview of the *skill-of-mind*-annotated dialogue dataset, and a detailed explanation of the evaluation criteria. This preparatory phase lasted approximately roughly 15 minutes.

Figure 6: A screenshot of human rating evaluation for MULTIFACETED SKILL-OF-MIND.

Figure 7: A screenshot of head-to-head comparison evaluation for DailyDialog Li et al. (2017)

| Template for Social Context Information in PROSOCIALDIALOGUE |
| --- |
| Speaker B should foster prosocial behavior by providing constructive feedback based on these Rule-of-Thumbs:\n- {rots} |
| Speaker B should encourage prosocial behavior by giving constructive feedback based on these Rule-of-Thumbs:\n- {rots} |
| To promote positive behavior, Speaker B should offer constructive feedback following these Rule-of-Thumbs:\n- {rots} |
| Guided by these Rule-of-Thumbs, Speaker B should encourage prosocial behavior through constructive feedback:\n- {rots} |
| Speaker B is expected to provide constructive feedback to encourage positive interactions, using these Rule-of-Thumbs:\n- {rots} |

Table 27: Template for social context information in PROSOCIALDIALOGUE Kim et al. (2022b). {rots} denotes Rule-of-Thumbs (RoTs).

| Template for Social Context Information in STARK (First Round Session) |
| --- |
| {name} is {age} years old, born in {birthplace}, and currently lives in {residence}. {event} |
| {name}, aged {age}, was born in {birthplace} and resides in {residence}. {event} |
| {name}, who is {age}, was born in {birthplace} and now lives in {residence}. {event} |
| {name} is {age}, originally from {birthplace}, and now living in {residence}. {event} |
| {name} is {age} years old, born in {birthplace}, and resides in {residence}. {event} |

Table 28: Template for social context information in STARK Lee et al. (2024d) (first round session).

| Template for Social Context Information in STARK (N-th Round Session) |
| --- |
| {name} is {age} years old, born in {birthplace}, and currently lives in {residence}. After {time_interval}, {name} has gone through {experience}, and now {event} |
| {name}, aged {age}, was born in {birthplace} and now resides in {residence}. Following {time_interval}, {name} experienced {experience}, and {event} |
| {name}, who is {age} years old, originally from {birthplace} and living in {residence}, went through {experience} after {time_interval}, and now {event} |
| {name} is {age}, born in {birthplace}, and currently resides in {residence}. After {time_interval} of {experience}, {name} has now {event} |
| {name}, {age} years old, from {birthplace} and residing in {residence}, has experienced {experience} over {time_interval}, and as a result, {event} |

Table 29: Template for social context information in STARK Lee et al. (2024d) (N-th round session).

| Template for Social Context Information in CACTUS |
| --- |
| Client's attitude is {client attitude}. The client's intake form is as follows:\n{client intake form}. |
| The client has an attitude of {client attitude}. Below is the client's intake form:\n{client intake form}. |
| With an attitude of {client attitude}, the client's intake form details are:\n{client intake form}. |
| Client's attitude: {client attitude}. Intake form information:\n{client intake form}. |
| The client's attitude is {client attitude}. Here is their intake form:\n{client intake form}. |

Table 30: Template for social context information in CACTUS Lee et al. (2024b).

## M    DISCUSSIONS

**Effect of Backbone LLMs.**    In this work, we primarily use the LLaMA series; however, as shown in Table 3, the Qwen-2.5 series achieves better performance. In future work, we will investigate whether training Qwen-2.5-Instruct-7B on MULTIFACETED SKILL-OF-MIND leads to a more effective skill-of-mind-infused LLM.

## N    PROMPT TEMPLATES

### N.1    A PROMPT TEMPLATE FOR SOCIAL CONTEXT INFORMATION

Table 27, Table 28, Table 29, Table 30, Table 31, Table 32, Table 33, Table 34, Table 35, Table 36 show social context template for PROSOCIALDIALOGUE Kim et al. (2022b), STARK Lee et al. (2024d) (first round session), STARK Lee et al. (2024d) (N-th round session), CACTUS Lee et al. (2024b), SYN-PERSONACHAT Jandaghi et al. (2023), CASINO Chawla et al. (2021) (sentence format), CASINO Chawla et al. (2021) (structured format), PEARL Kim et al. (2024a), PERSUASIONFORGOOD Wang et al. (2019), EMPATHETICDIALOGUES Rashkin (2018).

Template for Social Context Information in SYN-PERSONACHAT

User 1's Persona Information:\n- {user1 persona}\n\nUser 2's Persona Information:\n- {user2 persona}

User 1's Profile:\n- {user1 persona}\n\nUser 2's Profile:\n- {user2 persona}

Details of User 1's Persona:\n- {user1 persona}\n\nDetails of User 2's Persona:\n- {user2 persona}

Persona for User 1:\n- {user1 persona}\n\nPersona for User 2:\n- {user2 persona}

Information about User 1's Persona:\n- {user1 persona}\n\nInformation about User 2's Persona:\n- {user2 persona}

Table 31: Template for social context information in SYN-PERSONACHAT Jandaghi et al. (2023).

Template for social context information in CASINO (sentence format)

Speaker A is a {speaker_a_age}-year-old {speaker_a_ethnicity} {speaker_a_gender} who has a
{speaker_a_education} education. Their social value orientation is {speaker_a_svo}. According to the Big Five
personality traits, they score {speaker_a_extraversion} in extraversion, {speaker_a_agreeableness} in agreeableness,
{speaker_a_conscientiousness} in conscientiousness, {speaker_a_emotional_stability} in emotional stability, and
{speaker_a_openness_to_experiences} in openness to experiences. In the negotiation, Speaker A's highest priority
is {speaker_a_value2issue_high}, for which they reasoned: "{speaker_a_value2reason_high}". Their medium priority is
{speaker_a_value2issue_medium}, with the reasoning: "{speaker_a_value2reason_medium}". Their lowest priority is
{speaker_a_value2issue_low}, and they stated: "{speaker_a_value2reason_low}".
—
Speaker B is a {speaker_b_age}-year-old {speaker_b_ethnicity} {speaker_b_gender} who has a
{speaker_b_education} education. Their social value orientation is {speaker_b_svo}. Their Big Five personality traits
scores are {speaker_b_extraversion} in extraversion, {speaker_b_agreeableness} in agreeableness, {speaker_b_conscientiousness}
in conscientiousness, {speaker_b_emotional_stability} in emotional stability, and {speaker_b_openness_to_experiences} in openness
to experiences. During the negotiation, Speaker B's top priority is {speaker_b_value2issue_high}, and they explained:
"{speaker_b_value2reason_high}". Their medium priority is {speaker_b_value2issue_medium}, with the reason:
"{speaker_b_value2reason_medium}". Their lowest priority is {speaker_b_value2issue_low}, about which they mentioned:
"{speaker_b_value2reason_low}".

Table 32: Template for social context information in CASINO Chawla et al. (2021) (sentence format).

| Template for social context information in CASINO (structured format) |
| --- |
| Speaker A's Demographic Information: |
| - Age: {speaker_a_age} |
| - Gender: {speaker_a_gender} |
| - Ethnicity: {speaker_a_ethnicity} |
| - Education: {speaker_a_education} |
| Speaker A's Personality Information: |
| - Social Value Orientation (SVO): {speaker_a_svo} |
| - Big Five Personality Traits: |
|     - Extraversion: {speaker_a_extraversion} |
|     - Agreeableness: {speaker_a_agreeableness} |
|     - Conscientiousness: {speaker_a_conscientiousness} |
|     - Emotional Stability: {speaker_a_emotional_stability} |
|     - Openness to Experiences: {speaker_a_openness_to_experiences} |
| Speaker A's Negotiation Information: |
| - Priority Order (value2issue): |
|     - High: {speaker_a_value2issue_high} |
|     - Medium: {speaker_a_value2issue_medium} |
|     - Low: {speaker_a_value2issue_low} |
| - Personal Arguments (value2reason): |
|     - High: {speaker_a_value2reason_high} |
|     - Medium: {speaker_a_value2reason_medium} |
|     - Low: {speaker_a_value2reason_low} |
| — |
| Speaker B's Demographic Information: |
| - Age: {speaker_b_age} |
| - Gender: {speaker_b_gender} |
| - Ethnicity: {speaker_b_ethnicity} |
| - Education: {speaker_b_education} |
| Speaker B's Personality Information: |
| - Social Value Orientation (SVO): {speaker_b_svo} |
| - Big Five Personality Traits: |
|     - Extraversion: {speaker_b_extraversion} |
|     - Agreeableness: {speaker_b_agreeableness} |
|     - Conscientiousness: {speaker_b_conscientiousness} |
|     - Emotional Stability: {speaker_b_emotional_stability} |
|     - Openness to Experiences: {speaker_b_openness_to_experiences} |
| Speaker B's Negotiation Information: |
| - Priority Order (value2issue): |
|     - High: {speaker_b_value2issue_high} |
|     - Medium: {speaker_b_value2issue_medium} |
|     - Low: {speaker_b_value2issue_low} |
| - Personal Arguments (value2reason): |
|     - High: {speaker_b_value2reason_high} |
|     - Medium: {speaker_b_value2reason_medium} |
|     - Low: {speaker_b_value2reason_low} |

Table 33: Template for social context information in CASINO Chawla et al. (2021) (structured format).

---

Template for Social Context Information in PEARL

---

Seeker's overall movie preferences are represented as follows:\n{user persona}

Here is the seeker's complete movie profile:\n{user persona}

The seeker's general movie state is described below:\n{user persona}

Representation of seeker's overall movie interests:\n{user persona}

Below is the seeker's overall movie persona:\n{user persona}

---

Table 34: Template for social context information in PEARL Kim et al. (2024a).

---

Template for Social Context Information in PERSUASIONFORGOOD

---

Speaker A is attempting to persuade Speaker B.

In this scenario, Speaker A is the Persuader and Speaker B is the Persuadee.

Speaker A acts as Persuader, while Speaker B plays the role of Persuadee.

In the conversation, Speaker A is persuading Speaker B.

Speaker A aims to convince Speaker B.

---

Table 35: Template for social context information in PERSUASIONFORGOOD Wang et al. (2019).

---

Template for Social Context Information in EMPATHETICDIALOGUES

---

Speaker A is feeling {emotion} because {situation}.

Due to {situation}, Speaker A's emotion is {emotion}.

Speaker A's emotional state: {emotion}; Situation: {situation}.

Because of {situation}, Speaker A is in a {emotion} mood.

The situation is {situation}, so Speaker A feels {emotion}.

---

Table 36: Template for social context information in EMPATHETICDIALOGUES Rashkin (2018).

## N.2 A PROMPT TEMPLATE FOR MULTIFACETED SKILL-OF-MIND

**Prompt Template for *Skill-of-Mind* Generation**

**System Message:**

You are a helpful assistant that generates the most appropriate conversational skill and corresponding explanation. Read the provided instruction carefully.

---

**Instruction:**

In the given dialogue, two speakers are communicating with each other, and each speaker has their own information such as demographics, preferences, persona, current situation/narrative, past dialogue summaries, episodic memory, or other relevant details. This information is represented in the "[Social Context]" part. In this dialogue, image-sharing moments sometimes occur, represented in the format of "[Sharing Image] ¡image_description¿", where ¡image_description¿ represents the description of the shared image. You are also given the ideal response for the next turn in the given dialogue. Your task is to identify the most appropriate conversational skill that would lead to the ideal response in the given dialogue from the skill collection below, and explain why this particular skill was chosen. When generating the explanation, you should adopt the perspective of the speaker in the dialogue, selecting the skill based solely on the context of the given conversation. Do not consider the ideal response when generating your explanation; focus only on the given dialogue itself and why the chosen skill is the most suitable in that specific situation.

We provide the skill collection:
[Skill Collections]
- Empathy, Personal Background, Persona Recall, Self-disclosure, Negotiation, Conflict Resolution, Conflict Avoidance, Persuasion, Memory Recall, Topic Transition, Ethics, Harmlessness, Helpfulness, Avoiding Social Bias, Cultural Sensitivity, Commonsense Understanding, Rhetoric, Preference Elicitation, Knowledge Sharing, Knowledge Acquisition, Knowledge Searching, Active Listening, Factual Problem Solving, Logical Thinking, Critical Thinking, Creative Problem Solving, Immediate Response, Rephrasing, Echoing, Mentoring, Reflective Listening, Image-Sharing, Image-Commenting, Recommendation, Task Execution, Urgency Recognition, Clarification, Confirmation, Decision-making

Given the dialogue, social context information, and the next response, please brainstorm the most appropriate conversation skill and corresponding explanation.
[Social Context]
{social_context}

[Dialogue]
{dialogue}

[Next Response]
{response}

You should strictly follow the guidelines below:
[Guidelines]
- The answer should be represented in the form of a JSON list.
- Each entry in the list should be a Python dictionary containing the following keys: "skill", "explanation".
- The "skill" field should contain the one skill that is mostly required to generate the next response.
- The "explanation" field should provide a reason that occurs in the actual speaker's mind

before selecting the skill, from the speaker's perspective.
- The "explanation" should be written from the perspective of the actual speaker who made the next response.
- You can choose one or multiple skills if necessary, but each skill must have its own explanation.

[Generated Skills and Explanations ]

## N.3 A PROMPT TEMPLATE FOR SKILL-OF-MIND TASK

**Prompt Template for Skill-of-Mind Task**

You are provided with a dialogue between two speakers. Each speaker comes with additional information—such as demographics, preferences, persona, current situation or narrative, past dialogue summaries, episodic memory, and other relevant details—which is provided in the "[Social Context]" section.

[Social Context]:
{social_context}

[Dialogue]:
{dialogue}

Your task is to determine the most appropriate conversational skill that is relevant to and utilized in the given next response for "Speaker B", and explain why this particular skill was chosen. When generating the explanation, you should adopt the perspective of the speaker in the dialogue, justifying why this particular skill applies to the response. The selected skill should align with the dialogue and the provided context. Please select one of the following categories of conversational skills:
- {skill_categories}

Provide your final answer in the following JSON format:
json
{
"conversational_skill": ¡conversational_skill¿,
"explanation": ¡explanation¿
}

Answer:

**Prompt Template for Checklist-based Evaluation for Explanation Generation**

You will be provided with a previous dialogue history and two sentences, each representing the internal mind states of Speaker B before generating the next response. Your task is to evaluate the quality of these two sentences in terms of five aspects by answering "Yes" or "No" to each of the following criteria: Semantic Similarity, Perspective Consistency, Mentalizing, Non-Merging, and Interpretation Consistency. The questions for each aspect are described below.

### Checklist (Evaluation Items)
- Semantic Similarity: Do the two sentences convey essentially the same meaning (i.e., are they paraphrases or convey the same core content)?

- Perspective-Taking: Do both sentences consistently adopt Speaker A's perspective?
- Mentalizing (Avoiding Shortcut Pattern Matching): Do both sentences demonstrate an understanding of Speaker A's beliefs/thoughts/emotions in a way that goes beyond superficial copying of salient words (i.e., no obvious "shortcut" or purely pattern-based approach)?
- Non-Merging: Do both sentences avoid merging Speaker A's mental state with Speaker B's mental state (i.e., is it clear which beliefs, knowledge, or emotions belong to Speaker A alone)?
- Interpretation Consistency: Do both sentences describe or interpret Speaker A's current situation in a similar (non-contradictory) way?

### Dialogue:
{dialogue}

### Sentence A:
{sentence_a}

### Sentence B:
{sentence_b}

### Output Format:
```json
{
"Semantic Similarity": "¡Yes or No¿",
"Perspective-Taking": "¡Yes or No¿",
"Mentalizing": "¡Yes or No¿",
"Non-Merging": "¡Yes or No¿",
"Interpretation Consistency": "¡Yes or No¿"
}
```

### Guidelines:
- Compare the two given sentences based on the dialogue above for each of the five items in the checklist.
- Follow the output JSON format strictly. Do not include any additional explanations or descriptions.
- If either of the given two sentences is empty, respond with 'no' for all evaluation items.

Answer:

## N.4 A PROMPT TEMPLATE FOR RESPONSE GENERATION TASK

**Prompt Template for Response Generation (w/ Skill-of-Mind)**

Your task is to generate the most appropriate next response for "Speaker B" in a given dialogue between two speakers. You'll be given some social context about the two speakers of the dialogue, e.g., their relationship, demographic, preference, persona, or situation, etc.

When generating Speaker B's response, consider the provided explanation and conversational skill, but ignore them if they mislead the response. The explanation and conversational skill appear as Speaker B's internal thoughts, enclosed between ¡think¿ and ¡/think¿. Ensure the next response aligns with the style of the dialogue.

Only generate the most appropriate next response for "Speaker B" in the above dialogue, without any additional comments or descriptions.

[Social Context]
{social_context}

[Dialogue]
{dialogue}
¡think¿ {explanation} Thus, the most appropriate conversational skill for the next response is {skill}. ¡/think¿
Speaker B:

## N.5 PROMPT TEMPLATES FOR LLM-AS-A-JUDGE

**Prompt Template for LLM-as-a-Judge: Direct Assessment (w/ Reference Human Response)**

###Task Description:
An instruction (might include an Input inside it), a response to evaluate, a reference answer that gets a score of 5, and a score rubric representing a evaluation criteria are given.
1. Write a detailed feedback that assess the quality of the response strictly based on the given score rubric, not evaluating in general.
2. After writing a feedback, write a score that is an integer between 1 and 5. You should refer to the score rubric.
3. The output format should look as follows: "(write a feedback for criteria) [RESULT] (an integer number between 1 and 5)"
4. Please do not generate any other opening, closing, and explanations.

###The instruction to evaluate:
You will generate the next response in a dialogue (i.e., Speaker B's response) based on the provided social context. The dialogue is presented line by line, with each new line indicating a change in speaker.

[Social Context]
{social_context}

[Dialogue]
{dialogue_history}

###Response to evaluate:
{response}

###Reference Answer (Score 5):
{reference_answer}

###Score Rubrics:
{rubric}

###Feedback:

**Prompt Template for LLM-as-a-Judge: Pairwise Comparison (w/o Reference Human Response)**

###Task Description:
An instruction (might include an Input inside it), two responses to evaluate (denoted as Response A and Response B), a reference answer, and an evaluation criteria are given.

1. Write a detailed feedback that assess the quality of the two responses strictly based on the given evaluation criteria, not evaluating in general.
2. Make comparisons between Response A, Response B, and the Reference Answer. Instead of examining Response A and Response B separately, go straight to the point and mention about the commonalities and differences between them.
3. After writing the feedback, indicate the better response, either "A" or "B".
4. The output format should look as follows: "Feedback: (write a feedback for criteria) [RESULT] (Either "A" or "B")"
5. Please do not generate any other opening, closing, and explanations.

###The instruction to evaluate:
You will generate the next response in a dialogue (i.e., Speaker B's response) based on the provided social context. The dialogue is presented line by line, with each new line indicating a change in speaker.

###Response A:
{response_A}

###Response B:
{response_B}

###Score Rubrics:
{rubric}

###Feedback:

---

**Response Generation Scoring Rubric: Overall**

Overall, how effectively does the model's next-turn response in a multi-turn dialogue align with the context and address the user's needs?

Score 1: The response is severely off-topic or incoherent, showing no clear recognition of previous turns or the user's request.
Score 2: The response exhibits minimal awareness of context, referencing it only sporadically. It frequently diverges from the main topic or provides mostly irrelevant information.
Score 3: The response stays generally on track and offers some helpful content. However, it may lack depth, omit important details, or only partially address the user's needs.
Score 4: The response is coherent, context-aware, and effectively addresses the user's questions or needs. It demonstrates logical flow and includes relevant details without major errors or gaps.
Score 5: The response seamlessly integrates and synthesizes all relevant context from prior turns, delivering highly pertinent, precise, and helpful information that thoroughly meets the user's needs.

---

**Response Generation Scoring Rubric: Naturalness**

How natural does the model's next-turn response sound? Does it flow like typical human speech without awkward or robotic phrasing?

Score 1: The response sounds robotic, disjointed, or overtly "machine-like," with unnatural phrasing or syntax that disrupts comprehension.
Score 2: The response occasionally uses language or structures that feel forced or awkward; overall tone may be somewhat stiff.
Score 3: The response reads reasonably smoothly but may include moments of slightly off

or awkward phrasing. It generally flows without jarring language.

Score 4: The response is natural and fluid, demonstrating comfortable, human-like language. Minor stylistic oddities (if any) do not impede the overall natural feel.

Score 5: The response is exceptionally smooth and human-like, with a fluid, effortless tone and phrasing that feels entirely natural.

---

**Response Generation Scoring Rubric: Engagingness**

How engaging and captivating is the model's next-turn response? Does it spark genuine interest or encourage further interaction?

Score 1: The response is dull or uninteresting, showing little to no effort to engage the user. It may read like a lifeless statement.

Score 2: The response makes minimal effort to be engaging; it might have an occasional interesting point, but mostly feels flat.

Score 3: The response maintains a moderate level of interest—enough to hold the user's attention but without significant enthusiasm or color.

Score 4: The response is noticeably engaging, with lively or personable language. It demonstrates a clear attempt to foster a pleasant, interactive experience.

Score 5: The response is highly captivating, drawing the user in with vibrant, personable communication and genuinely encouraging further interaction.

---

**Response Generation Scoring Rubric: Consistency**

How consistently does the model's next-turn response align with the established context and persona? Does it avoid contradictions and maintain coherence with prior turns?

Score 1: The response clearly contradicts or ignores the established conversation context or persona, creating confusion.

Score 2: The response references the context in places but has notable inconsistencies — e.g., switching facts, tone, or persona mid-conversation.

Score 3: The response generally aligns with prior information and the speaker's established persona, with only minor or occasional inconsistencies.

Score 4: The response is strongly aligned with the conversation context, consistently maintaining facts, style, and persona from previous turns.

Score 5: The response impeccably maintains internal logic and persona across all turns, seamlessly integrating all conversation details without any contradictions.

---

**Response Generation Scoring Rubric: Specificity**

How specific and detailed is the model's next-turn response? Does it include precise, context-relevant information rather than vague or generic statements?

Score 1: The response is vague or generic, providing minimal or no actual detail relevant to the user's query or context.

Score 2: The response offers limited detail—there is some relevant information, but it remains mostly superficial or generic.

Score 3: The response includes moderately specific information, addressing some context points while still glossing over certain nuances.

Score 4: The response delivers substantial and well-targeted detail, addressing multiple relevant points in a thorough manner.

Score 5: The response is richly detailed and highly focused on the user's context, offering

precise, contextually tailored information.

