# OpenReview forum: "Enhancing Conversational Agents with Skill-of-Mind-Infused Large Language Model"
_ICLR.cc/2026/Conference — Submitted to ICLR 2026_

### Official Review · Reviewer_TvgS · 2025-10-29

**Soundness:** 3
**Presentation:** 4
**Contribution:** 3
**Rating:** 8
**Confidence:** 4

**Summary:**

This paper proposes Skill-of-Mind (SoM), a new idea for social dialogue that models how humans choose the right conversational skill—like empathy, persuasion, or humor—in conversation.
The authors build three parts:  1. a GPT-4–labeled dataset of 100k dialogues with 38 skills and explanations from 12 public sources. 2. THANOS models (1B/3B/8B): fine-tuned from LLaMA-3 to generate explanations and select skills. THANOSBENCH: a benchmark testing skill reasoning and response generation. Results show THANOS-8B outperforms LLaMA-3.1-70B/405B and nears Claude-3.5-Sonnet, while THANOS-3B enhances GPT-4o and Gemini responses without extra training.

**Strengths:**

- SoM is a fresh idea, different from chain-of-thought and commonsense reasoning. It focuses on strategy in social context.
- 100k labeled items covering 38 skills, with explanations + skills, which is rare and valuable for social context modeling.
- Fine-grained breakdown across abilities (Perspective-Taking, Semantic Similarity, Mentalizing, etc.) helps reveal current bottlenecks in social intelligence.

**Weaknesses:**

- The entire "skill-of-mind" annotation process for the dataset is based on automatic generation by GPT-4. Although the authors validated the annotation quality through human evaluation (Krippendorff's α = 0.62), this method may introduce GPT-4's own biases and limitations. The model's performance ceiling is, to some extent, limited by the capabilities of GPT-4 acting as the "teacher."
- SOM groundtruth are auto-generated by a single model. Even with some human checks, there’s a risk of closed-loop bias (GPT-4 labels → GPT-4 judges). It would be better to use independent judges (e.g., Claude) and more human evaluations.
- The paper lacks qualitative error analysis. It would benefit from showing representative failure cases—where the predicted skill mismatches the context or the generated explanation fails to justify the response—to clarify the limitations of the SoM reasoning process.

**Questions:**

- Have you performed ablation studies to quantify the individual contributions of the “explanation” and the “skill label”? It would also be helpful to clarify how much Skill-of-Mind improves the interpretability of the generated responses.
- I’m curious what the results would be if, instead of using the SoM skill as the intermediate explanation, GPT-4 directly generated other empathy-related reasoning strategies.
- How much did it cost to generate these datasets using GPT-4?

---

> ### Author Response · Authors · 2025-11-23
> **Response by Authors (1/3)**
>
> > **W1.** The entire "skill-of-mind" annotation process for the dataset is based on automatic generation by GPT-4. Although the authors validated the annotation quality through human evaluation (Krippendorff's α = 0.62), this method may introduce GPT-4's own biases and limitations. The model's performance ceiling is, to some extent, limited by the capabilities of GPT-4 acting as the "teacher."
>
> **A.** Yes, we also agree with the potential biases you mention, including stylistic bias. In the response-generation task of ThanosBench, no GPT-4o artifacts are present; the comparison involves model-generated responses versus purely human-generated responses (Table 5). Because the evaluator only judges appropriateness relative to human responses, this setting does not introduce GPT-4o-specific stylistic bias.
>
> To further mitigate this issue, one possible approach is to generate multiple skill-of-mind annotations for each sample using multiple teacher models—such as GPT, Claude, and Gemini. We believe that incorporating diverse teacher models in this way can help reduce stylistic bias.
>
> ---
>
> > **W2.** SOM groundtruth are auto-generated by a single model. Even with some human checks, there’s a risk of closed-loop bias (GPT-4 labels → GPT-4 judges). It would be better to use independent judges (e.g., Claude) and more human evaluations.
>
> **A.** We use additional evaluator LMs, including Claude-Sonnet-4 and LLaMA-3.1-70B-Instruct, to assess the skill-of-mind task in ThanosBench.
>
> | Models        | Evaluator LM             | Explanation Generation | Skill Classification | All    |
> |---------------|---------------------------|-------------------------|-----------------------|--------|
> | **Thanos-1B** | LLaMA-3.1-70B-Instruct   | 70.17                  | 30.71                | 14.62  |
> |               | GPT-4o-mini              | 51.40                  | 30.71                | 10.77  |
> |               | Claude-Sonnet-4          | 49.99                  | 30.71                | 3.31   |
> |               | **Avg.**                 | **57.19**              | **30.71**            | **9.57** |
> | **Thanos-3B** | LLaMA-3.1-70B-Instruct   | 74.82                  | 28.58                | 15.44  |
> |               | GPT-4o-mini              | 55.69                  | 28.58                | 10.95  |
> |               | Claude-Sonnet-4          | 54.08                  | 28.58                | 3.25   |
> |               | **Avg.**                 | **61.53**              | **28.58**            | **9.88** |
> | **Thanos-8B** | LLaMA-3.1-70B-Instruct   | 75.62                  | 33.14                | 17.93  |
> |               | GPT-4o-mini              | 59.48                  | 33.14                | 13.43  |
> |               | Claude-Sonnet-4          | 54.99                  | 33.14                | 3.67   |
> |               | **Avg.**                 | **63.36**              | **33.14**            | **11.68** |
> | **GPT-4o**    | LLaMA-3.1-70B-Instruct   | 79.21                  | 34.85                | 20.30  |
> |               | GPT-4o-mini              | 64.58                  | 34.85                | 17.34  |
> |               | Claude-Sonnet-4          | 50.15                  | 34.85                | 3.25   |
> |               | **Avg.**                 | **64.65**              | **34.85**            | **13.63** |
>
>
> As shown in the table, although the absolute values differ from Table 3, the performance pattern remains similar across evaluator LMs. Moreover, when evaluating responses generated by GPT-4o, LLaMA-3.1-70B-Instruct assigns higher scores, suggesting that there is likely no stylistic bias specific to GPT-4.
>
> We conducted an additional human evaluation on a randomly sampled set of 100 GPT-4o results (i.e., GPT-4o vs. GPT-4o + Thanos-3B) (same concern by reviewer wNvi).
>
> | Models              | Consistency | Engagingness | Naturalness | Specificity | Overall |
> |---------------------|-------------|--------------|-------------|-------------|---------|
> | GPT-4o              | 23.3        | 26.3         | 27.0        | 18.3        | 20.3    |
> | GPT-4o + Thanos-3B  | 76.7        | 73.7         | 73.0        | 81.7        | 79.7    |
>
> Even on GPT-4o responses, we observe clear improvements with Thanos-3B, suggesting that Thanos-3B is compatible with high-performing LLMs and can help them generate more human-preferred responses.

---

> > ### Comment · Reviewer_TvgS · 2025-11-27
> >
> > Thank you for the clarifications; some of my concerns were addressed. However, the issue with Q1 remains unresolved.

---

> ### Author Response · Authors · 2025-11-23
> **Response by Authors (2/3)**
>
> > **W3.** The paper lacks qualitative error analysis. It would benefit from showing representative failure cases—where the predicted skill mismatches the context or the generated explanation fails to justify the response—to clarify the limitations of the SoM reasoning process.
>
> **A.** We manually analyze 100 randomly sampled responses generated by GPT-4o. We identify three error categories: wrong skill-of-mind selection and overly verbose response.
>
> | Error Type                     | Ratio (%) |
> |--------------------------------|-----------|
> | Wrong Skill-of-Mind Selection  | 95        |
> | Overly Verbose Response          | 5        |
>
> From the table, we observe that wrong skill-of-mind selection constitutes the vast majority of errors. This suggests that selecting an appropriate skill-of-mind is crucial, as it directly influences the overall response quality. An example is presented as below:
>
> **Wrong Skill-of-Mind Selection**
> ```
> Dialogue:
> Speaker A: Room service.What can I do for you ?
> Speaker B: This is room 2012 . Where is my laundry ? You promised to send to me this morning .
> Speaker A: Well , let me check for you . I am sorry , sir . They mistook your order as a normal one.Actualty it's an express laundry , is that right ?
> Speaker B: Yes , I ordered the express service .
> Speaker A: Sir , we are terribly sorry for that.We will refund the charge and send your laundry to your room 3 hours later . Is that all right ?
>
>
> Predicted Skill-of-Mind:
> After being assured that my laundry is on its way and that the mistake has been rectified, I need to decide how best to proceed next. Since the issue with my laundry has been resolved to my satisfaction, I find it reasonable to ask if there is anything else I can assist with at this time. This shows readiness to move forward with the service and might also prompt any other queries or assistance I might need. Thus, the most appropriate conversational skill for the next response is Decision-making.
>
>
> GPT-4o Response:
> That's understandable, and thank you for apologizing and taking responsibility. I appreciate you offering to refund the charge and expedite the delivery. 3 hours should be okay for now.
>
> GPT-4o + Thanos-3B Response:
> That's all taken care of. Is there anything else I can assist you with?
> ```
> As shown above, an inappropriate skill-of-mind prediction leads to a suboptimal response compared to the baseline (GPT-4o) response.
>
>
> ---
>
> > **Q1.** Have you performed ablation studies to quantify the individual contributions of the “explanation” and the “skill label”? It would also be helpful to clarify how much Skill-of-Mind improves the interpretability of the generated responses.
>
> **A.** We conduct an ablation study on Skill-of-Mind by isolating the contributions of the “skill” and the “explanation” components. Specifically, we provide only the skill label or only the explanation to GPT-4o during response generation and compare the resulting responses against those generated with the full Skill-of-Mind signal.
>
> | Models                              | Overall | Naturalness | Engagingness | Specificity | Consistency |
> |-------------------------------------|---------|-------------|--------------|-------------|-------------|
> | GPT-4o + Thanos-3B                  | 51.27   | 53.48       | 54.11        | 52.22       | 53.48       |
> | vs. GPT-4o + Thanos-3B (only skill)     | 48.73   | 46.52       | 45.89        | 47.78       | 46.52       |
> | GPT-4o + Thanos-3B                  | 50.63   | 50.73       | 50.10        | 50.73       | 50.32       |
> | vs. GPT-4o + Thanos-3B (only explanation) | 49.37 | 49.27       | 49.90        | 49.27       | 49.68       |
>
> As shown in the table, using the full Skill-of-Mind signal yields the strongest performance. Notably, providing only the conversational skill is insufficient, especially when compared to the setting with only the explanation, highlighting that both components contribute meaningfully to response quality.

---

> ### Author Response · Authors · 2025-11-23
> **Response by Authors (3/3)**
>
> > **Q2.** I’m curious what the results would be if, instead of using the SoM skill as the intermediate explanation, GPT-4 directly generated other empathy-related reasoning strategies.
>
> **A.** As an empathy-related reasoning strategy, we adopt “perspective-taking aware zero-shot prompting” [2], which is specialized for empathetic dialogue generation and has been shown to be more effective than zero-shot CoT prompting. To apply this, we add “Let's put Speaker B in Speaker A's shoes and generate the most appropriate response.” to the input prompt during response generation on ThanosBench.
>
> | Models              | Overall | Naturalness | Engagingness | Specificity | Consistency |
> |---------------------|---------|-------------|--------------|-------------|-------------|
> | GPT-4o + Empathy Reasoning    | 46.33   | 33.01       | 38.54        | 49.37       | 48.25       |
> | GPT-4o + Thanos-3B  | 53.67   | 66.99       | 61.46        | 50.63       | 51.75       |
>
> As shown in the table, GPT-4o with Thanos-3B achieves better performance than empathetic reasoning. In particular, it shows substantial improvements in Naturalness and Engagingness. These results suggest that while empathy is helpful, it is not sufficient on its own in realistic scenarios. Instead, broader and more diverse conversational skills—captured by “skill-of-mind”—are necessary for enhanced social interaction.
>
> ---
>
> > **Q3.** How much did it cost to generate these datasets using GPT-4?
>
> **A.** We spent a total of $1,154 to generate 100K samples using the GPT-4 Batch API.
>
> ---
>
> References:
>
> [1] West, Peter, et al. "Symbolic knowledge distillation: from general language models to commonsense models." Proceedings of the 2022 conference of the North American chapter of the association for computational linguistics: Human language technologies. 2022.
>
> [2] Lee, Young-Jun, et al. "Investigating the effects of zero-shot chain-of-thought on empathetic dialogue generation." NeurIPS 2023 Workshop on Instruction Tuning and Instruction Following. 2023.

---

> ### Author Response · Authors · 2025-11-27
> **Response by Authors**
>
> We are glad to hear that, aside from Q1, your other concerns have been addressed. We understood your remaining concern (Q1) to be about the need for ablation studies that isolate the effect of providing only the skill or only the explanation in the prompt during response generation. In response, we presented comparative results for GPT-4o + Skill-of-Mind vs. GPT-4o + Skill, and GPT-4o + Skill-of-Mind vs. GPT-4o + Explanation.
>
> Would you kindly clarify which parts you still find unresolved and what kind of experimental design you had in mind? If you could let us know, we would be happy to conduct additional experiments to resolve your concern.

---

### Official Review · Reviewer_z354 · 2025-10-30

**Soundness:** 2
**Presentation:** 3
**Contribution:** 2
**Rating:** 4
**Confidence:** 4

**Summary:**

The paper introduces a novel concept called "skill-of-mind" to improve the social intelligence of LLMs in conversational settings. The core idea is to explicitly model the cognitive process of selecting an appropriate conversational skill by first generating a rationale ("explanation") and then choosing a skill from a predefined set. The authors create a large-scale dataset and use it to fine-tune a family of LLaMA-based models called THANOS for this task . The paper shows that using THANOS to guide other LLMs can significantly improve the quality of their generated responses, making them more natural and human-favorable. The work also contributes THANOSBENCH, a comprehensive benchmark for evaluating these capabilities .

**Strengths:**

1. The proposed "skill-of-mind" method is conceptually appealing and novel. It provides a structured and interpretable way to model a complex cognitive process by breaking it down into an explanation and a skill selection, moving beyond simple end-to-end response generation.
2. The experiments are extensive, consisting of fine-tuning and evaluation across 12 different base models from various families (GPT, Claude, Gemini, LLaMA, Gemma, Qwen) , which demonstrates the general applicability of the proposed method.
3. The topic is very interesting.

**Weaknesses:**

There is one main concern I have: the central claim of generating more "human-like" or "human-favorable" responses is not adequately supported by human evaluation, creating a methodological loop. The core evaluation of response quality (Table 5, 8) relies on another LLM (GPT-4o-mini) as the judge. The entire pipeline involves a dataset generated by GPT-4 (gpt-4-turbo) used to train a model (THANOS), which is then checklist-based  evaluated by another GPT model(GPT-4o-2024-11-20). This closed loop can only demonstrate that THANOS is effective at mimicking GPT-4's preferences, not that it is genuinely more human-like. The human evaluation presented (Table 6) is far too small (N=70 dialogues ) and limited to small models (Gemma-2-2B and LLaMA-3.1-8B ) to be considered conclusive evidence for the paper's primary claim.

**Questions:**

1. I’m finding it difficult to reconcile the paper's central claim of improving "human-like" responses with the heavy reliance on LLM-based evaluation. It seems that the pipeline, training on GPT-4's synthetic data and then evaluating with another GPT model, creates a methodological loop. I was wondering if the authors could provide a large-scale human A/B test, particularly for the stronger models like THANOS-guided GPT-4o, to more conclusively support the claims made in Table 5 and Table 8?
2. I'm still a bit unconvinced that the " explanation-skill-response " framework is fundamentally different from a well-crafted Chain-of-Thought (CoT) prompt. This  pipeline seems structurally analogous to CoT, and the comparison to DOCTOR in Table 8 doesn't quite address this, as it's a specialized model. I am very curious to know if the authors have results comparing their method to a generic CoT baseline (e.g., "think step-by-step before responding")? It would be interesting to see if the explicit "skill" component is truly necessary.
3. The list of 38 skills is quite extensive, but I found its structure a bit arbitrary, mixing high-level concepts like "Empathy" with simple behavioral descriptions like "Immediate Response." While I understand from Appendix D.2 that proposing a new taxonomy wasn't the goal, I was wondering if the authors have thought about the implications of this ad-hoc design? For instance, have you considered analyzing skill co-occurrence or grouping them into a more coherent hierarchy to provide a more principled justification for the list?
4. Could you clarify the multi-turn inference process? It appears the skill-of-mind reasoning is generated fresh each turn and then discarded, rather than being added to the conversational history. I am curious about the rationale for this design and whether an alternative, preserving a 'reasoning history', was considered.

---

> ### Author Response · Authors · 2025-11-23
> **Response by Authors (1/2)**
>
> > **W1 & Q1.** The central claim of generating more "human-like" or "human-favorable" responses is not adequately supported by human evaluation, creating a methodological loop.
>
> **A.** To address this concern, we conducted an additional human evaluation on a randomly sampled set of 100 GPT-4o results (i.e., GPT-4o vs. GPT-4o + Thanos-3B).
>
> | Models              | Consistency | Engagingness | Naturalness | Specificity | Overall |
> |---------------------|-------------|--------------|-------------|-------------|---------|
> | GPT-4o              | 23.3        | 26.3         | 27.0        | 18.3        | 20.3    |
> | GPT-4o + Thanos-3B  | 76.7        | 73.7         | 73.0        | 81.7        | 79.7    |
>
> Even on GPT-4o responses, we observe clear improvements with Thanos-3B, suggesting that Thanos-3B is compatible with high-performing LLMs and can help them generate more human-preferred responses.
>
> In addition, although we already reported meta-evaluation results in Appendix I, we conducted an additional meta-evaluation experiment directly on ThanosBench. Specifically, we asked several evaluator LLMs to select the more appropriate response (in terms of the overall metric) between two responses—one from GPT-4o and the other from GPT-4o + Thanos-3B—and measured the agreement with human judgments using accuracy. The meta-evaluation results on 100 randomly sampled responses are as follows:
>
> | Evaluator LLMs | Accuracy |
> |----------------|----------|
> | GPT-4o-mini    | 80       |
> | GPT-4o         | 84       |
>
> The agreement levels are sufficiently high, which suggests that the evaluator LLMs we used are reliable.
>
>
> ---
>
>
> > **Q2.** I'm still a bit unconvinced that the " explanation-skill-response " framework is fundamentally different from a well-crafted Chain-of-Thought (CoT) prompt. This pipeline seems structurally analogous to CoT, and the comparison to DOCTOR in Table 8 doesn't quite address this, as it's a specialized model. I am very curious to know if the authors have results comparing their method to a generic CoT baseline (e.g., "think step-by-step before responding")? It would be interesting to see if the explicit "skill" component is truly necessary.
>
> **A.** To show the effectiveness of the “skill-of-mind” concept, we present a pairwise comparison between GPT-4o with CoT reasoning and GPT-4o with Thanos-3B. For GPT-4o with CoT reasoning, we add “Please think step-by-step about what "Speaker B" would most likely say next. Then, provide only the response.” to the prompt template during response generation.
>
>
> | Models              | Overall | Naturalness | Engagingness | Specificity | Consistency |
> |---------------------|---------|-------------|--------------|-------------|-------------|
> | GPT-4o + CoT        | 48.73   | 35.14       | 36.19        | 49.84       | 48.42       |
> | GPT-4o + Thanos-3B  | 51.27   | 64.86       | 63.81        | 50.16       | 51.58       |
>
> As shown in the table, GPT-4o with Thanos-3B achieves better performance across all metrics. In particular, on Naturalness and Engagingness, GPT-4o with Thanos-3B shows a substantially larger improvement. For Specificity, the gap is not significant, which makes sense because CoT reasoning also encourages LLMs to generate specific responses. These results highlight that “skill-of-mind” is more appropriate for social reasoning.
>
> ---
>
> > **Q3.** The list of 38 skills is quite extensive, but I found its structure a bit arbitrary, mixing high-level concepts like "Empathy" with simple behavioral descriptions like "Immediate Response." While I understand from Appendix D.2 that proposing a new taxonomy wasn't the goal, I was wondering if the authors have thought about the implications of this ad-hoc design? For instance, have you considered analyzing skill co-occurrence or grouping them into a more coherent hierarchy to provide a more principled justification for the list?
>
> **A.** Yes, we thought about that before. However, we would like to say that, if we were proposing a new taxonomy for conversational skills, then employing a more comprehensive and rigorous approach—as you suggested—would certainly be necessary. However, since our skill list is designed purely for constructing a multifaceted skill-of-mind dataset, we focused on reviewing existing studies relevant to conversational AI and then discussed how to assign each skill into broader groups. In addition, as mentioned in Lines 815–818, our skill list is meant to provide clarity, and each skill is not strictly bound to a single group; therefore, we did not pursue a more principled justification. We would like to emphasize that while developing a fully principled taxonomy would indeed be helpful, for the scope of this work, we believe that the approach we took is sufficient.

---

> ### Author Response · Authors · 2025-11-23
> **Response by Authors (2/2)**
>
> > **Q4.** Could you clarify the multi-turn inference process? It appears the skill-of-mind reasoning is generated fresh each turn and then discarded, rather than being added to the conversational history. I am curious about the rationale for this design and whether an alternative, preserving a 'reasoning history', was considered.
>
> **A.** We agree that extending skill-of-mind reasoning to a more interactive and dynamic environment—where reasoning history is preserved and evolves over turns—is necessary and represents a promising direction for future work. We may also consider leveraging recent research on reasoning memory [1]. Thank you again for highlighting this.
>
> ---
>
> References:
>
> [1] Ouyang, Siru, et al. "Reasoningbank: Scaling agent self-evolving with reasoning memory." arXiv preprint arXiv:2509.25140 (2025).

---

### Official Review · Reviewer_W2NB · 2025-10-31

**Soundness:** 3
**Presentation:** 3
**Contribution:** 3
**Rating:** 4
**Confidence:** 4

**Summary:**

This paper introduces skill-of-mind, a novel framework inspired by how people talk and think in social settings. The main idea is that before replying, an AI should first decide what kind of skill is needed, like empathy, clarification, or ethics. and after that generate its answer. They curate a dataset with 100k samples. Each sample contains explanations and 38 conversation skills taken from 12 sources. These also include info like who is speaking, their relationship, and memories. They then used this dataset to train a group of models, ranging in size, to guess both the reasoning and the skill behind what someone says. They tested their models on a test set with 8 datasets to check how well models do in skill reasoning and how good their responses sound, using both humans and LLMs as judges. The main finding is that even the best models like GPT-4o and Claude-3.5 perform poorly on skill-of-mind tasks. But adding their trained 3B model, makes LLMs sound more natural, empathetic, and human-like.

**Strengths:**

- The concept of skill-of-mind, which connects human social thinking, theory of mind, and real conversation strategy, is interesting.
- The proposed dataset is strong as it also contains explanations with the skill labels.
- THANOSBENCH tests not only if a model can guess the right skill, but also if it helps make better replies. The checklist-style evaluation makes results more clear and fair.
- The author's proposed system worked better than existing SOTA methods to produce more natural replies.
- The paper is well written and is easy to follow.

**Weaknesses:**

- The training and testing methodology is creating a loop in terms of LLM dependance. That is, the skills and explanations come from GPT-4, and then THANOS is trained and tested on that same GPT-4 data. This way, the benchmark might just reward models that copy GPT-4's style instead of showing real human-like thinking.
- The 38 skills mix different kinds of things like memory, social behavior, and how people move topics. Although this mix is practical, but it is also confusing. It’s hard to tell what exactly the model is learning, like is it learning a mental skill, a social act, or a speaking trick.
- The results show broad (coarse) skills work better than fine-grained ones, but the paper doesn’t explain why. Maybe small skill differences are too subjective or noisy. Not exploring this further feels like a missed chance.

**Questions:**

- Real talk often needs more than one skill at once. Can we model uncertainty or mixing like "empathy + clarification" instead of forcing one skill per turn?
- Why does Immediate Response make things worse (Figure 5)? Maybe it shows thinking before speaking is more important than speed in social talk.
- Can we personalize skill-of-mind? For example, pick different skills for introverts vs extroverts, or adapt over time for each user.

---

> ### Author Response · Authors · 2025-11-23
> **Response by Authors (1/2)**
>
> > **W1.** The training and testing methodology is creating a loop in terms of LLM dependance. That is, the skills and explanations come from GPT-4, and then THANOS is trained and tested on that same GPT-4 data. This way, the benchmark might just reward models that copy GPT-4's style instead of showing real human-like thinking.
>
> **A.** In the response-generation task of ThanosBench, no GPT-4o artifacts are present; the comparison involves model-generated responses versus purely human-generated responses (Table 5). Because the evaluator only judges appropriateness relative to human responses, this setting does not introduce GPT-4o-specific stylistic bias.
>
> ---
>
> > **W2.** The 38 skills mix different kinds of things like memory, social behavior, and how people move topics. Although this mix is practical, but it is also confusing. It’s hard to tell what exactly the model is learning, like is it learning a mental skill, a social act, or a speaking trick.
>
> **A.** Thank you for raising this point. We agree that it can be difficult for a model to determine which specific skill it should apply during learning. This is precisely why we incorporate social context information —it serves as conditional information or a contextual hint to model.. For example, in Table 11, the social context information includes the keyword “prosocial behavior.” With this cue, we expect the model to more naturally consider prosocial-behavior–related skills.
>
> ---
>
> > **W3.** The results show broad (coarse) skills work better than fine-grained ones, but the paper doesn’t explain why. Maybe small skill differences are too subjective or noisy. Not exploring this further feels like a missed chance.
>
> **A.** We interpret the phenomenon as follows: coarse-grained skill-of-mind performs better than fine-grained skill-of-mind. In Table 21, the gap in the ExG task is not large (fine is slightly higher by about +3%), but in the SC task, the difference becomes substantial, with coarse-grained skills showing a +25.8% gain. The reason is the mismatch between the ThanosBench skill distribution and the fine-grained skill-of-mind categories. For example, as mentioned in Lines 1290–1293 of Appendix H.5, ThanosBench and the coarse-grained skill set share the same conversational-skill category. Both only cover “Empathy,” while the fine-grained version specifically targets “Empathetic Inquiry,” which introduces a more detailed requirement. We believe this mismatch leads to the performance gap. Nonetheless, we also believe that handling fine-grained skill-of-mind for more practical and realistic scenarios is an important and promising direction for future work.

---

> ### Author Response · Authors · 2025-11-23
> **Response by Authors (2/2)**
>
> > **Q1.** Real talk often needs more than one skill at once. Can we model uncertainty or mixing like "empathy + clarification" instead of forcing one skill per turn?
>
> **A.** Yes, this is possible. Beyond building a dialogue generation model that first thinks about the skill-of-mind and then generates a response simultaneously (Section 6.3, Lines 472–475), modeling mixed or combined skills-of-mind is indeed a promising direction for more realistic scenarios, since humans often rely on multiple skills at the same time. we believe that future work should expand the dataset so that each dialogue turn contains explicit annotations for multiple skill-of-mind.
>
> ---
>
> > **Q2.** Why does Immediate Response make things worse (Figure 5)? Maybe it shows thinking before speaking is more important than speed in social talk.
>
> A. Since ThanosBench evaluates empathetic responding, dialogue safety, and related behaviors, the model needs to generate responses carefully while considering the interlocutor’s mind and situation. This is why simply “speaking faster” does not work well. Therefore, we agree with your point that “thinking before speaking” is important in the social reasoning setting.
>
> ---
>
> > **Q3.** Can we personalize skill-of-mind? For example, pick different skills for introverts vs extroverts, or adapt over time for each user.
>
> A. Yes, personalizing skill-of-mind is definitely a promising future direction. We also believe that personalization is an important factor for building more socially interactive models. In our work, we incorporate “social context information,” which provides broad social cues and can include persona-related information. This work does not directly address personalized skill-of-mind but we will add this discussion about the previous studies for personalization as follows:, constructing personalized skill-of-mind capabilities would be very valuable—for example, using Big-5 Chat [1] as seed data. Additionally, similar to long-term dialogue studies [2], we can consider experience-based personalization that adapts over time using memory systems [3,4].
>
> ---
>
> References:
>
> [1] Li, Wenkai, et al. "Big5-chat: Shaping llm personalities through training on human-grounded data." Proceedings of the 63rd Annual Meeting of the Association for Computational Linguistics (Volume 1: Long Papers). 2025.
>
> [2] Lee, Young-Jun, et al. "Stark: Social long-term multi-modal conversation with persona commonsense knowledge." Findings of the Association for Computational Linguistics: EMNLP 2024. 2024.
>
> [3] Maharana, Adyasha, et al. "Evaluating very long-term conversational memory of llm agents." arXiv preprint arXiv:2402.17753 (2024).
>
> [4] Ouyang, Siru, et al. "Reasoningbank: Scaling agent self-evolving with reasoning memory." arXiv preprint arXiv:2509.25140 (2025).

---

### Official Review · Reviewer_wNvi · 2025-11-01

**Soundness:** 2
**Presentation:** 3
**Contribution:** 3
**Rating:** 4
**Confidence:** 4

**Summary:**

This paper introduces the concept of Skill-of-Mind (SoM), to enhance the human-likeness and naturalness of LLM responses in complex social dialogues. The central hypothesis is that providing LLMs with appropriate social skill guidance can effectively improve their response quality.
The primary contributions of this work are:
1. The MULTIFACETED SKILL-OF-MIND dataset, a corpus of nearly 100k samples annotating the conversational skills (from a summarized set of 38) and the explaination.
2. THANOS, a series of models trained on this dataset, designed to provide social skill decisions.
3. The THANOSBENCH benchmark, developed to evaluate both SoM capabilities and response generation quality.
The authors' experiments demonstrate that: (1) The THANOS models outperform baseline models of equivalent and even larger scales on SoM tasks. (2) Using the SoM (including explanations and skills) generated by THANOS to guide other LLMs significantly enhances the naturalness and engagingness of their responses .

**Strengths:**

1. **Large-scale, comprehensive dataset**: The paper constructs a large-scale SoM-annotated dataset with 100k samples and 38 skill categories. This is one of the largest and most skill-comprehensive datasets in this domain, making it a valuable resource for research on social dialogue reasoning.
2. **Holistic benchmark**: THANOSBENCH provides a comprehensive evaluation framework that assesses not only the final response quality but also the intermediate "Skill-of-Mind" reasoning process, including explanation generation (ExG) and skill classification (SC) . This is a crucial step toward evaluating interpretable and controllable dialogue systems.
3. **Relatively strong empirical results**: The experiments show that the THANOS models, specifically infused with SoM capabilities, perform robustly. Moreover, THANOS-3B, when used as a guidance mechanism, significantly improves the social response quality of various SOTA LLMs, including GPT-4o , demonstrating the method's practicality and effectiveness.

**Weaknesses:**

1. **Heavy reliance on LLM-as-a-judge with inadequate justification**: The evaluation in THANOSBENCH relies heavily on an LLM-as-a-judge. In Appendix I, the authors attempt to validate the consistency between the LLM Evaluator and humans in other multi-turn social dialogue settings, but the correlation is not high (e.g., a Spearman correlation of 0.489 ). It is widely recognized (e.g., in work on the SOTOPIA benchmark) that LLMs as judges can exhibit various biases, especially when directly comparing two responses in a complex, subjective setting (like the RESPONSE GENERATION TASK). The small-scale human validation in the main paper , conducted only on two smaller models (Gemma-2-2B and LLaMA-3.1-8B), is insufficient to substantiate the broad claims made about performance improvements on SOTA models like GPT-4o.
2. **Questionable inter-rater agreement**: When assessing the dataset's annotation quality, the authors report an inter-rater agreement using Krippendorff’s α of 0.62, claiming this "indicates a substantial level of agreement". However, this value is often considered moderate at best. According to established guidelines (e.g., doi:10.1016/j.mex.2023.102545), an alpha < 0.67 may suggest poor agreement among raters. This raises concerns about the reliability of human annotation.
3. **Lack of ablation studies**: A key argument of the paper is that accurate SoM reasoning and classification prompts improve response quality. However, the authors do not sufficiently verify the inverse: whether providing an incorrect or suboptimal skill and its explaination would relatively degrade response quality. For instance, an ablation could involve using the SoM outputs from weaker models (from Table 3) to guide the SOTA models (in Table 5). The experiment in Appendix H.6, while related, is insufficient as it only injects the skill label without the corresponding explaination , making the setup inconsistent with the main experiments. This is a critical omission, as the presence of a longer, chain-of-thought-style explanation—regardless of its accuracy—may in itself contribute to improved response quality.

**Questions:**

1. Can the authors provide more evidence for the reliability of the LLM-judge, particularly for the RESPONSE GENERATION TASK, where subjective comparison is required?
2. In the human evaluation presented in Table 6, why were only these two smaller models (Gemma-2-2B and LLaMA-3.1-8B) selected for assessment? Why not include larger, SOTA models to provide a more comprehensive and compelling validation of the THANOS mechanism's effectiveness?
3. Have the authors considered ablation studies to verify the effect of different strategies for providing the SoM (skill + explanation) on response quality? Specifically, why not use the better-performing models from Table 3 (e.g., GPT-4o or Qwen-2.5-72B) to provide the SoM guidance in the response generation experiments, rather than THANOS-3B?
4. Have the authors considered testing the latest LLMs or LRMs (Large Reasoning Models) on these tasks? It would be valuable to see if explicit SoM prompting still holds an advantage over, for instance, a general long reasoning strategy.
5. A minor presentation issue: In the prompt templates in Appendix N, there are several instances of text like "¡think¿" and "¡Yes or No¿" . This might be a rendering error.

---

> ### Author Response · Authors · 2025-11-23
> **Response by Authors (1/2)**
>
> > **W1 & Q1 & Q2.** Insufficiently validating the LLM-as-a-judge’s reliability—especially for response generation on larger model—and lacking sufficient human evaluation on larger model
>
> **A.** We agree that an LLM evaluator can be subjective for open-ended dialogue generation tasks. To address this concern, we conducted an additional human evaluation on a randomly sampled set of 100 GPT-4o results (i.e., GPT-4o vs. GPT-4o + Thanos-3B).
>
> | Models              | Consistency | Engagingness | Naturalness | Specificity | Overall |
> |---------------------|-------------|--------------|-------------|-------------|---------|
> | GPT-4o              | 23.3        | 26.3         | 27.0        | 18.3        | 20.3    |
> | GPT-4o + Thanos-3B  | 76.7        | 73.7         | 73.0        | 81.7        | 79.7    |
>
> Even on GPT-4o responses, we observe clear improvements with Thanos-3B, suggesting that Thanos-3B is compatible with high-performing LLMs and can help them generate more human-preferred responses.
>
> In addition, although we already reported meta-evaluation results in Appendix I, we conducted an additional meta-evaluation experiment directly on ThanosBench. Specifically, we asked several evaluator LLMs to select the more appropriate response (in terms of the overall metric) between two responses—one from GPT-4o and the other from GPT-4o + Thanos-3B—and measured the agreement with human judgments using accuracy. The meta-evaluation results on 100 randomly sampled responses are as follows:
>
> | Evaluator LLMs | Accuracy |
> |----------------|----------|
> | GPT-4o-mini    | 80       |
> | GPT-4o         | 84       |
>
> The agreement levels are sufficiently high, which suggests that the evaluator LLMs we used are reliable.
>
>
> ---
>
> > **W2.** Questionable inter-rater agreement: When assessing the dataset's annotation quality, the authors report an inter-rater agreement using Krippendorff’s α of 0.62, claiming this "indicates a substantial level of agreement". However, this value is often considered moderate at best. According to established guidelines (e.g., doi:10.1016/j.mex.2023.102545), an alpha < 0.67 may suggest poor agreement among raters. This raises concerns about the reliability of human annotation.
>
> **A.** Based on the reference you provided, we have revised the statement from “indicates a substantial level of agreement” to “indicates a moderate level of agreement” in our updated PDF.
>
> ---
>
> > **W3.** Ablation studies: whether weaker SoM model guide SOTA response generaion model.
>
> **A.** We conduct an ablation study to examine whether Skill-of-Mind generated by a weaker model (LLaMA-3.1-8B-Instruct) can guide GPT-4o to produce better responses.
>
> | Models                          | Overall | Naturalness | Engagingness | Specificity | Consistency |
> |---------------------------------|---------|-------------|--------------|-------------|-------------|
> | GPT-4o                          | 59.81   | 60.76       | 58.23        | 59.49       | 61.08       |
> | GPT-4o + LLaMA-3.1-8B-Instruct  | 40.19   | 39.24       | 41.77        | 40.51       | 38.92       |
>
> As shown in the table, weaker Skill-of-Mind outputs do not help GPT-4o generate better responses. This suggests that a sufficient level of Skill-of-Mind capability is necessary to provide effective guidance.

---

> ### Author Response · Authors · 2025-11-23
> **Response by Authors (2/2)**
>
> > **Q3.** Have the authors considered ablation studies to verify the effect of different strategies for providing the SoM (skill + explanation) on response quality? Specifically, why not use the better-performing models from Table 3 (e.g., GPT-4o or Qwen-2.5-72B) to provide the SoM guidance in the response generation experiments, rather than THANOS-3B?
>
> **A.** We would like to emphasize that the main purpose of this paper is to demonstrate the effectiveness of the Thanos model in the response generation task. Therefore, we initially focused on employing Thanos-3B as the SoM generator. However, based on your helpful comment, we additionally conducted ablation experiments using stronger models as SoM providers. The results are shown below:
>
> | Models                       | Overall | Naturalness | Engagingness | Specificity | Consistency |
> |------------------------------|---------|-------------|--------------|-------------|-------------|
> | GPT-4o                       | 35.44   | 36.22       | 33.12        | 31.53       | 37.34       |
> | GPT-4o + Thanos-3B           | 64.56   | 63.78       | 66.88        | 68.47       | 62.66       |
> | GPT-4o                       | 44.73   | 44.68       | 42.15        | -455.63     | 48.16       |
> | GPT-4o + GPT-4o              | 55.27   | 55.32       | 57.85        | 555.63      | 51.84       |
> | GPT-4o                       | 45.63   | 44.37       | 43.10        | 47.22       | 48.48       |
> | GPT-4o + Qwen2.5-72B-Instruct| 54.37   | 55.63       | 56.90        | 52.78       | 51.52       |
>
> Notably, even though GPT-4o and Qwen-2.5-72B-Instruct achieve higher performance than Thanos-3B on the skill-of-mind task (Table 3), Thanos-3B actually makes GPT-4o generate more human-preferred responses compared to using GPT-4o or Qwen-2.5-72B-Instruct as the SoM generator.
>
> We believe this is because the response generation task in ThanosBench involves broader and more complex scenarios—such as commonsense-driven conversations (MuTual) and empathetic responding—beyond the narrower skill-of-mind classification setting. In these more diverse and open-ended conversational situations, Thanos-3B provides more effective and actionable SoM guidance than general-purpose LLMs, resulting in higher downstream response quality.
>
> ---
>
> > **Q4.** Have the authors considered testing the latest LLMs or LRMs (Large Reasoning Models) on these tasks? It would be valuable to see if explicit SoM prompting still holds an advantage over, for instance, a general long reasoning strategy.
>
> **A.** We evaluate recent LLMs/LRMs—including Claude-Sonnet-4.5, GPT-OSS-120B, and GPT-4.1—on the skill-of-mind task in ThanosBench. For the reasoning models, the default reasoning effort and maximum token limit are set to medium (only for the OpenAI series) and 10K, respectively. The top-p and temperature are set to 0.9 and 1.0.
>
> | Models             | Explanation Generation | Skill Classification | All    |
> |--------------------|-------------------------|-----------------------|--------|
> | GPT-4o             | 64.58                   | 34.85                 | 17.34  |
> | Claude-Sonnet-4.5  | 64.12                   | 32.49                 | 19.23  |
> | GPT-OSS-120B       | 74.30                   | 31.07                 | 22.31  |
> | GPT-4.1 | 72.05 | 35.15 | 23.91 |
>
> As shown in the table, GPT-4.1 achieves the best performance among all models. Among LRMs, Claude-Sonnet-4.5 and GPT-OSS-120B outperform GPT-4o, which was the SOTA model reported in the main paper (Table 3).
>
> Moreover, we explore whether a test-time scaling (TTS) strategy (i.e., sequential longer thinking) improves performance on the skill-of-mind task. Specifically, we set the reasoning effort to high and evaluate GPT-OSS-120B.
>
> | Models                 | Explanation Generation | Skill Classification | All   |
> |------------------------|-------------------------|-----------------------|-------|
> | GPT-OSS-120B (medium)  | 74.30                   | 31.07                 | 22.31 |
> | GPT-OSS-120B (high)    | 73.23                   | 32.66                 | 22.37 |
>
> Simply applying longer thinking does not substantially improve performance on the skill-of-mind task. For example, on the Explanation Generation task, TTS decreases performance, whereas it slightly improves the Skill Classification task. These findings suggest that longer thinking alone is not always beneficial; rather, effectiveness depends on the task domain. For skill-of-mind tasks, specialized strategies beyond extended reasoning are required.
>
> ---
>
> > **Q5.** A minor presentation issue: In the prompt templates in Appendix N, there are several instances of text like "<think>" and "<Yes or No>" . This might be a rendering error.
>
> **A.** Thank you for pointing this out. We have corrected these rendering issues in the revised PDF.

---

### Meta-Review · Area_Chair_P8ze · 2026-01-04

**Summary:**

The paper proposes "Skill-of-Mind" (SoM), a framework to help LLMs choose conversational skills like empathy or persuasion. While the reviewers found the concept interesting and the 100K dataset scale impressive, there are major concerns about the "methodological loop" where GPT-4 is used for both training and evaluation. The data quality is also a sticking point, with a low inter-rater agreement ($\alpha = 0.62$). Despite the authors' rebuttal and extra human evaluations, the core issues regarding the arbitrary skill taxonomy and the lack of theoretical depth remain unresolved.

**Reviewer Concerns:**

Addressed in Rebuttal: The authors provided additional human evaluation on 100 GPT-4o samples to show the SoM effectiveness. They also included a comparison with standard Chain-of-Thought (CoT) prompting to show that SoM performs better in social engagement.

Not Addressed: The dataset quality is still a primary issue; the inter-rater agreement is only "moderate" at best, which is risky for a benchmark-focused paper. The skill taxonomy (38 skills) is poorly justified, mixing high-level social concepts with basic behaviors. Furthermore, Reviewer TvgS explicitly noted that the ablation study to isolate the contribution of the "explanation" vs. "skill label" is still not convincing.

**Reviewer Scores:**

Reviewer wNvi: 4. Won't move much because the low alpha score is a dealbreaker for a benchmark paper.

Reviewer W2NB: 4. Might stay low as the "GPT-4 loop" concern is still there.

Reviewer z354: 4. Likely no change. The author's answer on skill taxonomy was too weak.

Reviewer TvgS: 8 -> 6. They directly said the rebuttal didn't fix their concerns on the ablation logic.

---

### Decision · Program_Chairs · 2026-01-26

Reject